

# 1 Predicting carbon dioxide and energy fluxes across global
# 2 FLUXNET sites with regression algorithms

Gianluca Tramontana[1], Martin Jung[2], Gustau Camps-Valls[3], Kazuhito Ichii[4,5], Botond Raduly[1,6], Markus
Reichstein[2], Christopher R. Schwalm[7], M. Altaf Arain[8], Alessandro Cescatti[9], Gerard Kiely[10], Lutz
Merbold[11], Penelope Serrano-Ortiz[12], Sven Sickert[13], Sebastian Wolf[14] and Dario Papale[1].
[1]Department for Innovation in Biological, Agro-food and Forest systems (DIBAF), Univeristy of Tuscia, Viterbo, 01100,
Italy,
[2]Max Planck Institute for Biogeochemistry, Jena, 07745, Germany;
[3]Image Processing Laboratory (IPL), Paterna (València), 46980, Spain.
[4]Department of Environmental Geochemical Cycle Research, Japan Agency for Marine-Earth Science and Technology,
Yokohama, 236-0001, Japan.
[5]Center for Global Environmental Research, National Institute for Environmental Studies, Tsukuba, 305-8506, Japan.
[6]Department of Bioengineering Sapientia Hungarian University of Transylvania, Miercurea Ciuc, 530104, Romania.
[7]Woods Hole Research Center, Falmouth MA, 02540, USA.
[8]School of Geography and Earth Sciences, McMaster University, Hamilton (Ontario), L8S4L8, Canada
[9]Institute European Commission, Joint Research Centre, Institute for Environment and Sustainability, Ispra, Ispra, 21027,
Italy
[10]Civil & Environmental Engineering and Environmental Research Institute, University College, Cork, T12 YN60, Ireland.
[11]Department of Environmental Systems Science, Institute of Agricultural Sciences, ETH Zurich, Zurich, 8092, Switzerland.
[12]Department of Ecology, University of Granada, Granada, 18071, Spain.
[13]Computer Vision Group, Friedrich Schiller University Jena, 07743 Jena, Germany
[14]Department of Environmental Systems Science, ETH Zurich, Zurich, 8092, Switzerland.
*Correspondence to:* G. Tramontana (g.tramontana@unitus.it)
**Abstract.** Spatial-temporal fields of land-atmosphere fluxes derived from data-driven models can complement simulations
by process-based Land Surface Models. While a number of strategies for empirical models with eddy covariance flux data
have been applied, a systematic intercomparison of these methods has been missing so far. In this study, we perform a cross-
validation experiment for predicting carbon dioxide ($CO_2$), latent heat, sensible heat and net radiation fluxes, in different
ecosystem types with eleven machine learning (ML) methods from four different classes (kernel methods, neural network,
tree methods, and regression splines). We employ two complementary setups: (1) eight days average fluxes based on
remotely sensed data, and (2) daily mean fluxes based on meteorological data and mean seasonal cycle of remotely sensed
variables. The pattern of predictions from different ML and setups were very consistent. There were systematic differences
in performance among the fluxes, with the following ascending order: net ecosystem exchange ($R^2<0.5$), ecosystem
respiration ($R^2>0.6$), gross primary production ($R^2>0.7$), latent heat ($R^2>0.7$), sensible heat ($R^2>0.7$), net radiation ($R^2>0.8$).
ML methods predicted very well the across sites variability and the seasonal cycle ($R^2> 0.7$) of the observed fluxes, while the
eight days deviations from the mean seasonal cycle were not well predicted ($R^2< 0.5$). Fluxes were better predicted at
forested and temperate climate sites than at ones growing in extreme climates or less represented in training data (e.g. the
tropics). The large ensemble of ML based models evaluated will be the basis of new global flux products.
*Keywords*: Machine learning, carbon fluxes, energy fluxes, FLUXNET, remote sensing, FLUXCOM

## 39 1. Introduction

Improving our knowledge of the carbon, water, and energy exchanges between terrestrial ecosystems and the atmosphere is
essential to better understand and model Earth's climate system (IPCC, 2007; Reich, 2010). In situ continuous observations



can be obtained by the eddy covariance technique, which estimates the net exchanges of carbon dioxide ($CO_2$), water vapour and energy between land ecosystems and the atmosphere (Aubinet at al., 2012; Baldocchi et al., 2014). Under ideal conditions eddy covariance estimations are equals to the vertical turbulent fluxes plus the changes in the scalar storage under the measurement height, while under the non-ideal conditions advection needs to be accounted (Aubinet et al., 2005, Hammerle et al., 2007).

The large-scale measurement network, FLUXNET integrates site observations of these fluxes globally and provides detailed time series of carbon and energy fluxes across biomes and climates (Baldocchi et al., 2008). However, eddy-covariance measurements are site-level observations, and spatial upscaling is required to estimate these fluxes at regional to global scales.

The increasing number of eddy covariance sites across the globe has encouraged the application of data-driven models by machine learning (ML) methods such as Artificial Neural Networks (ANNs, Papale et al., 2003), Random Forest (RF, Tramontana et al., 2015), Model Trees (MTE, Jung et al., 2009; Xiao et al., 2008, 2010) or Support Vector Regression (SVR, Yang et al., 2006, 2007) to estimate surface-atmosphere fluxes from site level to regional or global scales (e.g. Beer et al., 2010, Jung et al., 2010, 2011; Kondo et al., 2015; Schwalm et al., 2010; Yang et al., 2007; Xiao et al., 2008, 2010). The ML upscaled outputs are also increasingly used to evaluate land surface models (e.g., Anav et al., 2013; Bonan et al., 2011; Ichii et al., 2009; Piao et al., 2013).

The key characteristic of data-driven models compared the process-based ones are the former's intrinsic observational nature, and that functional relationships are not prescribed but rather emerge from the patterns found in the measurements. In this context, data-driven models extract multivariate functional relationships between the in situ measured fluxes of the network and explanatory variables in an empirical way. The explanatory variables are generally coming from satellite remote sensing, providing information on vegetation state (e.g., vegetation indices) and other land surface properties (e.g., surface temperature), along with continuous measurements of meteorological variables at flux towers.

While ML-based upscaling provides a systematic approach to move from point-based flux estimates to spatially explicit gridded fields, various sources of uncertainty exist. For example, individual ML methods might have different responses especially when these models are applied beyond the conditions sampled by the training dataset (Jung et al., 2009; Papale et al., 2015). The information content of the driving input variables may not be sufficient to capture the variability of the fluxes in all conditions (Tramontana et al., 2015). Moreover, remotely sensed and meteorological gridded datasets are affected by uncertainties themselves. Remote sensing data contain noise, biases and gaps, and can be perturbed by atmospheric effects or by the presence of snow. Meteorological gridded datasets are known to contain product specific biases (Garnaud et al., 2014; Tramontana et al., 2015; Zhao et al., 2012).

Thorough experiments using multiple data-driven models and explanatory variables are an essential step to identify and assess limitations and sources of uncertainty in the empirical upscaling approach. In this study, we present and evaluate an ensemble of ML based empirical models to predict $CO_2$ and energy fluxes across FLUXNET sites. The participating models were selected in the context of the FLUXCOM initiative, forming the basis of subsequent global flux products. We performed a consistent cross-validation for two complementary experimental setups using: (1) eight days average fluxes based on remotely sensed data, and (2) daily mean fluxes based on remotely sensed and meteorological data. The ML tools have spanned the full range of commonly applied algorithms: from model tree ensembles, multiple adaptive regression splines, artificial neural networks, to kernel methods, with several representatives of each family. The different ML algorithms were trained with consistent sets of predictor variables. Our overarching aim was to understand how well fluxes of $CO_2$ (gross primary production (GPP), terrestrial ecosystem respiration (TER) and net ecosystem exchange (NEE)), and energy (latent heat (LH), sensible heat (H) and net radiation (Rn)) could be predicted by an ensemble of ML methods. More specifically, we addressed the following questions:

   1.   Were the patterns of predicted fluxes consistent between the two experimental setups?




2.  How different were the predictions of the various ML algorithms?

3.  How did the performance differ among capturing the across-sites, seasonal and the deviations from the mean seasonal cycle variability?

4.  How did the performance differ among climate zones or ecosystem types?

**2. Material and methods**

**2.1 Data**

**2.1.1 Eddy covariance study sites**

We used eddy covariance data from 224 flux-tower sites (supplementary material, Sect. S1), which originate from the FLUXNET La Thuile synthesis dataset and CarboAfrica network (Valentini et al., 2014). The study sites were distributed globally and cover most plant functional types (PFT) and biomes over the globe (Table 1, Giri et al., 2005).

**2.1.2 Observation-based $CO_2$ and energy fluxes**

All flux measurements were post-processed using standardized procedures of quality control and gap-filled following Reichstein et al. (2005) and Papale et al. (2006). Estimates of GPP and TER were derived from half-hourly NEE measurements using two independent flux partitioning methods: (1) following Reichstein et al. (2005), where the temperature sensitivity of ecosystem respiration was initially estimated from night-time NEE data and then extrapolated to daytime to estimate TER and GPP, by subtracting NEE (negatively signed for the $CO_2$ uptake) from TER; (2) following Lasslop et al. (2010), where daytime NEE data were used to constrain an hyperbolic light response curve to directly estimate GPP and TER. In the following we reference GPP and TER derived by Reichstein et al. (2005) as $GPP_R$ and $TER_R$; whereas estimates based on the Lasslop et al. (2010) method are referred to as $GPP_L$ and $TER_L$.

The half-hourly data were aggregated to daily values and screened according to multiple quality criteria, as follow:

1) we excluded data when more than 20% of the data were based on gap-filling with low confidence (Reichstein et al., 2005);

2) we identified and removed obviously erroneous periods due to non-flagged instrument or flux partitioning failures based on visual interpretation;

3) we excluded data-points where the two flux-partitioning methods provided extremely different patterns. Specifically, we computed for each site a robust linear regression between (a) $TER_R - GPP_L$ and NEE, and (b) $GPP_R$ and $GPP_L$. Data points with a residual outside the range of ± 3 times of the inter-quartile range were removed. This criterion removed only the extreme residuals, systematic differences between methods were not removed;

4) we removed the 5% of data-points with the largest friction velocity (u*) uncertainty, defined as data points above the 95[th] percentile of daily u* uncertainty (measured as the inter-quartile range of 100 bootstrap samples).

Similarly to the $CO_2$ fluxes, we applied the same criteria 1) and 2) for the energy fluxes. Additionally, we removed data with inconsistent energy fluxes, i.e. when the residual of a robust linear regression between LE + H and Rn for each site was outside three-times the inter-quartile range of the residuals.

**2.1.2 Remote sensing data**

We collected data coming from Moderate Resolution Imaging Spectroradiometer (MODIS) which provided data at a spatial resolution of 1km or better (Justice et al., 2002). We used the cutout surroundings of 3×3 km pixels centered on the towers. We collected and processed the following MODIS products: MOD11A2 Land Surface Temperature (LST; Wan et al., 2002),




MOD13A2 Vegetation Index (Normalized Difference Vegetation Index (NDVI) and Enhanced Vegetation Index (EVI);
Huete et al., 2002), MOD15A2 Leaf Area Index (LAI) and Fraction of Absorbed Photosynthetic Active Radiation (FPAR;
Myneni et al., 2002), and MCD43A2 and MCD43A4 Bidirectional Reflectance Distribution Function (BRDF) corrected
surface reflectances (Schaaf et al., 2002). The BRDF-corrected surface reflectance data were further processed to calculate
the Normalized Difference Water Index (NDWI) (Gao, 1996) and the Land Surface Water Index (LSWI) (Xiao et al., 2002).
These data were obtained from http://daac.ornl.gov/MODIS/.
The remote sensing data were further processed to improve data quality and data gaps were filled to create continuous time-
series data, and to minimize non-land surface signals. We adopted the following processing scheme: we identified good
quality pixels by the using the quality assurance/quality criteria (QA/QC) included in the MODIS product. If more than 25%
of the pixels had good quality at the time of snapshot, the average of good quality pixels were assigned as the actual value.
Otherwise, the data at the time snapshot were marked as blank (no data). Then, we created the mean seasonal variations from
2000-2012 using only good pixels data and the data gaps in the processed data were filled using the mean seasonal variation.
Only MOD13 was provided with 16 days composite, and eight days data were created by assigning the 16 days composite
value to the corresponding two eight days periods.
**2.1.3 Meteorological data**
The in situ measured air temperature (Tair), global radiation (Rg), VPD, and precipitation were used after data screening
according to the criteria 1) and 2) as for the measured fluxes (see Sect. 2.1.2). We also used long-term time series of these
variables from ERA-Interim for the period 1989-2010 (Dee et al., 2011), which were bias-corrected for each site on the basis
of the overlapping period of in situ measurements (see http://www.bgc-jena.mpg.de/~MDIwork/meteo/). These long-term
meteorological data were primarily used to calculate consistent metrics of climatological variables (e.g. mean annual
temperature) across all sites given the temporal coverage of data of the different sites. In addition, we used a composite of
these ERA-Interim data based and in situ measured data to obtain a gap-free time series for calculating a simple soil Water
Availability Index (WAI, see Sect. 2.3.2 and supplementary material, Sect. S3).
**2.2 Participating ML methods**
For our purpose, 11 ML algorithms for regression from four broad families were chosen: tree-based methods, regression
splines, neural networks, kernel methods. Moreover a comprehensive review of ML algorithms in biophysical parameter
estimation can be found in Verrelst et al. (2015). At follow a brief description of the characteristics of each family.
Tree based methods
These methods construct hierarchical binary decision trees. The inner nodes of the tree hold decision rules according to
explanatory variables (e.g. less/greater than X1), recursively splitting the data into subspaces. The leaf nodes at the end of
the decision tree contain models for the response variable. Because a single tree is generally not effective enough to cope
with strong non-linear multivariate relationships, ensembles of trees are often used. We applied two different tree ensemble
methods: (1) Random Forests (RF) which combines regression trees grown from different bootstrap samples and randomly
selected features at each split node (Breiman, 2001; Ho, 1998); and (2) Model Tree Ensembles (MTE) which combine model
trees (Jung et al., 2009). The main difference between regression and model trees is the prediction model in the leaf node: a
simple mean of the target values from the training in regression trees and a parametric function (here a multiple linear
regression) in model trees. In this study, we used three different variants of MTE, which differ mainly with respect to
different cost functions for determining the splits, and the technique to create the ensemble of model trees. Further details are
described in the supplementary material (Sect. S2).
Regression splines



Multivariate regression splines (MARS) are an extension of simple linear regression adapted to non-linear response surfaces
using piecewise (local) functions. The target variable is predicted as the sum of regression splines and a constant value
(Alonso Fernández, 2013; Friedman et al., 1991).
Neural networks
Neural networks are based on nonlinear and nonparametric regressions. Their base unit is the neuron, where nonlinear
regression functions are applied. The neurons are interconnected and organized in layers. The output of $m$ neurons in the
current layer are the inputs for $n$ neurons of the next layer. We used two types of neural networks: the artificial neural
network (ANN) and the group of method for data handle (GMDH). In an ANN, each neuron performs a linear regression
followed by a non-linear function. Neurons of different layers are interconnected by weights that are adjusted during the
training (Haykin et al., 1999; Papale et al., 2003). The GMDH is a self-organizing inductive method (Ungaro et al., 2005)
building polynomials of polynomials; the neurons are pairwise connected through a quadratic polynomial to produce new
neurons in the next layer (Shirmohammadi et al., 2015).
Kernel methods
Kernel methods (Shawe-Taylor and Cristianini, 2004) owe their name to the use of kernel functions, which measures
similarities between input data examples. Among the available kernel methods we used: (1) support vector regression (SVR)
(Vapnik et al., 1997), (2) kernel ridge regression (KRR) (Shawe-Taylor and Cristianini, 2004), and (3) Gaussian process
regression (GPR) (Rasmussen, 2006; Verrelst et al., 2012). The SVR defines a linear prediction model over mapped samples
to a much higher dimensional space, which is non-linearly related to the original input (Verrelst et al., 2012, Yang et al.,
2007). The KRR is considered as the kernel version of the regularized least squares linear regression (Shawe-Taylor and
Cristianini, 2004, Verrelst et al., 2012). The GPR is a probabilistic approximation to nonparametric kernel-based regression,
and both a predictive mean (point-wise estimates) and predictive variance (error bars for the predictions) can be derived. We
also used a hybrid approach combining RF with simple decision stumps in the inner nodes and GPR for prediction in the leaf
nodes (Fröhlich et al., 2012).
**2.3 Experimental design**
**2.3.1. Experiment setups**
We defined two complementary experimental setups, which differ in the choice of explanatory variables, and the temporal
resolution of the target fluxes: 1) at eight days temporal resolution using exclusively remote sensing data (hereafter RS); and
2) at daily temporal resolution using meteorological data together with the mean seasonal cycle (MSC) of remote sensing
data (hereafter RS+METEO). In the latter case, the MSC of remote sensing data were smoothed and interpolated to daily
time step. Each setup was characterized by advantages and disadvantages. While RS could provide products with high
spatial resolution (up to 1km), data were limited to the MODIS era (2000-present) and had a coarse temporal resolution. The
uncertainties of remote sensing data at tower locations, due to finer scale spatial heterogeneity, also degraded the
performance of the ML methods. RS+METEO could take advantage of information from meteorological variables, and was
resistant to the noise of remote sensing time series because only mean seasonal cycle of data from satellite were used.
RS+METEO allowed for upscaled products over a longer time period (because not constrained by the availability of MODIS
data) and finer time scale (daily). However the predictive skill of this setting was conditioned by the missing of information
regarding the interannual variability of vegetation greenness. In addition the use of meteorological gridded datasets
introduced another source of uncertainty coming from potential dataset specific biases and by their typically coarse spatial
resolution (0.5 degrees or larger).
**2.3.2. Variable selection**



Combining remote sensing and meteorological data (see Sect. 2.1.2 and 2.1.3) we created additional variables for model
inputs. In the case of RS+METEO setup we derived the Water Availability Index (WAI) that it was based on a simple soil
water balance model (for more details see supplementary material, Sect. S3) as an attempt to better represent water stressed
conditions. For both setups we derived proxies for absorbed radiation as the product between vegetation greenness (e.g. EVI,
NDVI, FPAR) and drivers related to the absorbable energy for photosynthesis (e.g. daytime LST, Rg, and potential
radiation). Other derived variables included the mean seasonal cycle (MSC) of dynamic variables and associated metrics
(minimum, maximum, amplitude, and mean). For remote sensing predictors, the MSC and associated metrics were based on
the period 2001-2012 while for climate variables were based on the bias corrected daily long-term ERA-Interim data
reference period (1989-2010). In total, 216 potential explanatory variables were created for RS and 231 for RS+METEO (see
supplementary material S4 for details).
For each setup we selected a small subset of variables best suited to predict the target fluxes using a variable selection search
algorithm. Variable selection was an important component in the spatial upscaling because the accuracy of predictions
improves and the computational costs of the global predictions were minimized. We used the Guided Hybrid Genetic
Algorithm (GHGA) published by Jung and Zscheischler (2013), which was designed for variable selection problems with
many candidate predictor variables and computationally expensive cost functions. The GHGA required the training of a
regression algorithm (here RF) to estimate the cost associated with selected variable subsets (see S5 for details).
Instead of doing a computationally demanding variable selection for each individual flux, variable selection runs were
performed for the RS and RS+METEO setups and separately for $CO_2$ and energy fluxes. This procedure had the advantage
that the resulting global products originated from a consistent set of predictor variables. All trained machine learning used
the same drivers for predictions and they were listed in Table 2.

### 2.3.3. Model training

The capability of ML methods to spatially extrapolate $CO_2$ and energy fluxes was evaluated by a 10-fold cross-validation
strategy. The training datasets were stratified into 10-folds, each one roughly containing 10% of the data. Entire sites were
assigned to each fold (Jung et al., 2011). The training of machine learning was done using data of nine folds while the
prediction was done for the remaining one. This operation was repeated 10 times and each one of the 10 fold was used
exactly once as validation set. In this way: a) the validation sets were completely independent from the training sets, b) the
validation of the models was done to the entire dataset. Due to computational expense of the RS+METEO setup, only one
method representing each "family" – multiple regressions, RF, MARS, ANN and KRR – was trained.
ML methods base settings were tuned before the training (for further details, see supplementary material S6). These hyper-
parameters accounted for regularization in order to avoid overfitting, as well as for the shape and smoothness constraints.
Instead, the model parameters were estimated for each ML every time in each fold

### 2.3.4. Model evaluation

In order to highlight the differences between the RS and RS+METEO setups, the daily output from RS+METEO were
aggregated to eight days time steps; the same periods and sites were used for the comparison. Besides the statistical analysis
of the individual ML cross-validation results, we focused on the ensemble median estimate, here defined as the median
predicted value across all ML for a given setup and time step. The advantages of the median ensemble estimate was the
robustness of the predictions for the contribution of many ML that reduced the risk of outlier in the extrapolation exercise.
We used different metrics to evaluate the ML performance such as the Nash and Sutcliffe model efficiency (MEF), the root
mean square error (MSE), the empirical BIAS, the Pearson's linear correlation coefficient (ρ), the coefficient of
determination ($R^2$) and the ratio of variance (ROV).




MEF (Nash and Sutcliffe, 1970) was the capability of a model to estimate a target variables better than a reference model. If
the reference model was the mean value of the observation, MEF was calculated as:
$$MEF = 1 - \frac{\sum_{i=1}^{n}(x_i - y_i)^2}{\sum_{i=1}^{n}(y_i - \overline{y})^2} \qquad (1)$$

where $x_i$ and $y_i$ were the predicted and the observed values respectively and $\overline{y}$ is the mean value of the observations. MEF
varied between -inf to 1; in the case of MEF > 0 the predictive skill of the model was better than the mean (MEF = 1 for the
ideal model), instead if MEF=0 the predictive skill of the model was equivalent to the mean, finally if MEF < 0, the
predictive skill of the mean value of the target was better than the model.
The RMSE was estimates as the root square of the mean value of the squared residuals:
$$RMSE = \sqrt{\frac{\sum_{i=1}^{n}(x_i - y_i)^2}{n}} \qquad (2)$$

The BIAS was evaluated as the mean value of model's residuals
$$BIAS = \frac{\sum_{i=1}^{n}(x_i - y_{i)}}{n} \qquad (3)$$

Following Gupta et al. (2009) the importance of bias on the overall uncertainty was evaluated as the ratio between the square
of BIAS and the Mean Square Error, the latter estimated as the square value of RMSE.
The Pearson's linear correlation coefficient (ρ) was the ratio between the covariance between the modeled and observed
values ($\sigma_{xy}$) and the product of the standard deviation of modeled ($\sigma_x$) and observed ($\sigma_y$) values:
$$\rho = \frac{\sigma_{xy}}{\sigma_x \sigma_y} \qquad (4)$$

$R^2$ was estimated as the squared value of ρ; finally ROV was evaluated as the ratio between predicted and observed standard
deviation.
We evaluated the overall predictive skill of the models, evaluating the consistency among trained ML approaches and across
the experimental setup. Then we evaluated the capability of the regression models to predict site-specific mean fluxes, mean
seasonal cycle (MSC), and anomalies (Jung et al., 2011). The MSC per site was calculated using the averaged values for
each eight days period across all available years, but only when at least two values (i.e., years) for each eight days period
were available. To assess the mean values of the study sites, we calculated the mean of the MSC if at least 50% of the 46
eight days values were present, whereas the eight days anomalies were calculated as the deviation of a flux value from the
MSC. Finally, the mean site value was removed from the MSC to disentangle the seasonal variation from the mean site
values, making them as complementary.
We also analyzed the performance for the different Köppen climate zone and IGBP plant functional types. In particular. we
computed for each flux, setup and tower site the performances of ML median estimate. Then, for each setup, we estimated
the median value of the site-by-site statistics per PFT and climate zone.
**3. Results and Discussions**





### 3.1 Overview

The ensemble median estimate always outperformed the median performance of ML-specific methods (the median value of metrics calculated for individual ML) (Table 3; Appendix A). Individual ML methods also exhibited higher skill than multiple linear regressions (higher MEF and lower RMSE; Fig. 1). This highlighted the added value of ML methods as these were able to account for nonlinearities in either explanatory variables or fluxes. Overall, using the ensemble median estimate gave a representative overview of ML-based flux predictions.

### 3.2 Predictive skill of $CO_2$ and energy fluxes

Predictive skill of the ensemble median estimate clustered into tiers whereby energy fluxes were uniformly better predicted than $CO_2$ fluxes: Rn > H/LE/GPP > TER > NEE (Table 3). The highest skill levels as exhibited by net radiation showed near perfect agreement; Rn displayed a model efficiency (MEF) of 0.91-0.92 and a correlation of 0.96. The decline in skill for the second tier fluxes was ca. 15% to 20%; MEF for H, LE, and GPP is 0.79, 0.75-0.76, and 0.71 respectively. The lowest two tiers exhibited 20% and 40% declines in MEF (0.57-0.64 and 0.43-0.46 for TER and NEE respectively). These relative rankings were unchanged regardless of skill metric used—apart from RMSE where the difference in fluxes units and magnitude, confounded a direct comparison (Table 3)—suggesting that accuracy and precision scale linearly.

There were only minor performance differences between the two $CO_2$ fluxes partitioning methods (Table 3), although for the RS setup, the performance of $TER_L$ were comparatively lower than $TER_R$ (lower MEF, ρ and ROV). A similar trend was not found in the case of RS+METEO setup.

The overall skill profile in this study confirmed previous upscaling efforts (Jung et al., 2011; Yuan et al., 2010). This relatively stable cross-study skill gradient reflected the information content of the available predictor variables. The spatiotemporal variability of remotely sensed land surface properties was well-suited to predict the top tier fluxes (Rn, H, LE, and GPP) (Jung et al., 2008; Tramontana et al., 2015; Xiao et al., 2010;.Yang et al., 2007)

The higher skill associated with energy fluxes suggested that these fluxes were more easily predictable using the drivers selected in particular respect to NEE. In fact NEE was strongly controlled by external factors such management and disturbances (Amiro et al., 2010; Thornton et al., 2002) and by lag and memory effects (Bell et al., 2012; Frank et al., 2015; Papale et al., 2015; Paruelo et al., 2005), which were both poorly captured by predictor variables typically used in upscaling and poorly constrained in general, i.e., data limited. Another reason for the low performances in NEE simulation was in the random uncertainties of the measurements that were larger compared to H and LE and had an important effect being NEE the difference between two large components (GPP and TER).

Among the $CO_2$ fluxes, GPP was the best predicted probably because the seasonal cycle and canopy properties, which were strongly related to GPP, were well represented by the ML drivers. The intermediate skill of TER, relative to $CO_2$ fluxes only, was supported by its tight coupling to the well-predicted GPP and the availability of predictor variables that captured the temperature dependency of respiration. However specific drivers for TER could be still missing. In fact in contrast to GPP, the canopy properties were less important drivers of TER, while the soil properties, carbon pools and their turnover rates were key for respiratory processes (Amiro et al., 2010) but not available to be used as drivers. This likely explains the poor performance for TER in comparison with GPP.

### 3.3 Were the flux predictions consistent between RS and RS+METEO?

Skills, in terms of both performance tiers and absolute value of skill metrics, were similar for both RS and RS+METEO approaches with some differences, in particular: (1) RS and RS+METEO diverged more for those fluxes where they showed lower overall skill levels, in particular the NEE (Fig. 1, Table 3); (2) MEF and correlation values were slightly larger for RS than RS+METEO, excluding $TER_L$ where the opposite was found, indicating an important role of the meteorological data for



this version of the ecosystem respiration. It should be considered that the differences in performances could be also due to a
different ensemble size, with the RS composed of 11 individual ML-based ensemble members, whereas RS+METEO was
based on only four. The overall good performance of the RS setup implied that $CO_2$ and energy fluxes can be mapped
exclusively based on remotely sensed inputs allowing high-spatial resolution products and reduction of uncertainty due to the
meteorological drivers spatialization (Tramontana et al., 2015).
Nonetheless, the differences between the experimental setups were less appreciable.

**3.4 How different were the predictions of the various ML algorithms?**

Pair-wise $R^2$ values for model outputs (Table 4) were close to unity ($R^2 \geq 0.90$), regardless of experimental setup, with NEE
showing a slightly lower value ($R^2 = 0.84$). Among corresponding model residuals (Table 4), $R^2$ values ranged from 0.79
(Rn) to 0.89 ($TER_L$). Comparing the same ML technique but using different experimental setups (Table 4, RS vs.
RS+METEO) showed similarly high, albeit somewhat diminished level of consistency ($R^2$ range ranged from 0.71 to 0.80
for model residuals). These finding highlighted that the ML methods were mapping between explanatory variables and target
fluxes both reliably and robustly. Across the all three consistency checks there was also a tendency for better predicted
fluxes (e.g., H) to exhibit higher pair-wise $R^2$ values than poorly predicted fluxes (e.g., NEE). This was expected as more
robust patterns—and therefore those that lead to greater predictive skill—were easier to extract regardless of ML algorithm
and experimental setup in this study; thus increasing consistency. While this broad consistency confirmed that the extracted
patterns were robust, the decline in $R^2$ when comparing the same ML trained with different drivers (RS vs RS+METEO)
respect to the correlation among ML methods with the same drivers, suggested that the choice of the explanatory variable
had higher impact than the choice of the ML technique for the pattern of predictions.

**3.5 How did the performance differ among capturing the across-sites, seasonal and the deviations from the mean seasonal cycle variability?**

Decomposing FLUXNET data into across-sites variability, mean seasonal cycle, and interannual variability components
(Sect. 2.3.4) revealed clear gradients in predictive skill (Table 5 and Fig. 2). Across-sites variability was in general well-
captured by the ML ($R^2$ range: 0.61 to 0.81 except for NEE) and the best predicted pattern were found for GPP and TER.
This suggests that ML were suitable to reproduce the spatial pattern of mean annual fluxes.
The variability in the mean seasonal cycle (at eight days time scale) was also uniformly well predicted ($R^2$ between 0.67-0.77
for GPP and TER, and between 0.86-0.98 for the energy fluxes) and the best predicted pattern for energy fluxes in particular
for LE and Rn.
The importance of seasonality in $CO_2$ and energy annual fluxes was known (Joiner et al., 2014; Jung et al., 2011; Merbold et
al., 2009; Wolf et al., 2011) and biases in its dynamic (e.g. in the growing season length) could lead to biases into the
predicted fluxes (Ichii et al., 2010). A clear benefit of the ML upscaling approach used here was that none of the parameters
controlling seasonality were prescribed, reducing the possibility of biases.
In contrast, the eight days anomalies variability were generally poorly captured by all the ML approaches used with only H
and Rn showing an $R^2$ greater than 0.4. This low predictive skill held regardless of whether eight days, monthly (Jung et al.,
2011), or annual time steps were used (data not shown). This was likely due to a combination of missing predictor variables
(e.g. disturbances, management, legacy effects) and the noise/uncertainty in both predictor and target variables that played a
major role when small differences (like in the interannual variability) were predicted. The slightly better performances when
sensible heat flux was estimated could be due to the lower uncertainty in this flux respect to the others (only one sensor used,
the sonic anemometer, in contract with the other fluxes where also the gas analyzer was used) but also to the fact that it was




strongly and directly related to the LST used as driver. In any case, predicting interannual variability remained one of the
largest challenges in the context of empirical upscaling.
NEE was confirmed to be the most difficult and consequently poorest predicted flux (Table 3). NEE showed considerably
lower skill relative to the other fluxes for across-sites variability ($R^2 = 0.46$), the mean seasonal cycle ($R^2 = 0.59$), and
interannual variability ($R^2 = 0.13$, $TER_L$ was the lowest at 0.10).
**3.6 How did the performance differ among climate zones or ecosystem types?**
Using climate zone and plant functional type (PFT) to disaggregate ML methods performances we found that in general
energy fluxes were better predicted than $CO_2$ fluxes among the different climate zone and PFTs (Fig. 3 and Appendix C).
The median $R^2$ between simulation and observation for $CO_2$ fluxes (excluding NEE) was greater than 0.6 for more than 75%
of the PFT and climate zone, while in the energy fluxes the $R^2$ greater was higher than 0.7 for more than the 85% of the PFT
and climate zone (in all sites for Rn).
NEE was again consistently poorly predicted (low $R^2$ and high relative RMSE; Fig. 3), apart from deciduous broadleaf
forests (DBF) and MF where a marginal improvement was evident. The better performance in these two vegetation types
could be related to the higher seasonal variance of NEE in comparison with the other PFTs that was a pattern more
consistent with the seasonal variance of the used drivers.
Overall the ML methods showed poor prediction capability in the tropics and in the evergreen broadleaf forests (EBF).
Possible explanations were the absence of a clear seasonal cycle traceable by the remote sensing signal (evergreen
vegetation) and a low variance in their seasonal cycle that was challenging to explain mathematically and capture with a
model. (Sims et al., 2008; Yebra et al., 2015; Yuan et al., 2010). In addition, the difficulty in acquiring cloud-free remotely
sensed data, from one introduced additional uncertainty in the drivers.
Reason for low performances could be also the lack of important driving variables, that were probably the main explanation
for cropland, where management information were missing (e.g. irrigation, fertilization, tillage) and also for cold and dry
environments where the extreme nature of the environments characterized by water limitation (for dry sites) and the extreme
low temperature (for cold sites) required more targeted drivers such direct estimations of soil water content. In addition, cold
and dry sites were characterized by both low magnitude and low variance of fluxes, making difficult to explain the fluxes
variability in these ecosystems types by empirical models.
**4. Conclusions**
The ML methods presented and evaluated in this study have shown high capability to predict $CO_2$ and energy fluxes, in
particular the between-site variability and the seasonal variations, with a general tendency of increasing performance in the
following order: NEE, TER, GPP, LE, H, and Rn. The relatively poor performance for NEE likely resulted from factors that
cannot be easily accounted for in ML based modelling approaches, such as legacies of site history (e.g. disturbances,
management, age and stocks). Future progress in this direction requires the reconstruction of the relevant management and
disturbance history, trying to integrate information from forest inventories, high resolution satellites such LANDSAT and
high resolution biomass data from radar and LIDAR with the aim to improve model performance. The better results obtained
for the energy fluxes (LE and H) in comparison to the $CO_2$ fluxes (GPP and TER) could be related to more complex
mechanisms driving the carbon cycle that were also not represented in the drivers used, in addition to a relatively higher
uncertainties in the GPP and TER due to the use of flux partitioning methods based on NEE measurements.
We found no substantial bias in the predictions of the ML models for most vegetation types or biomes. However, the
predictions have deviated more from the observations for evergreen broadleaf forests, croplands, the tropics and extreme
climates. The growing number of eddy-covariance sites, in particular new sites in poorly represented regions will likely



improves the predictive skill in the future. This is particularly relevant since tropical areas account for a disproportionate
share of the global water and carbon cycle (Beer et al., 2010).
The deviations from the mean seasonal cycle (eight days anomalies) are still poorly captured by the ML methods. We
expected the RS+METEO approach to perform better regarding the prediction of anomalies as meteorological drivers were
included and the noise in remote sensing time series was greatly reduced by using smoothed seasonal cycle. However, this
was not the case, which indicates that either, the eight days anomalies of both, the flux data, and the drivers, were strongly
affected by the uncertainties, or that the anomalies were dominated by management and disturbance (or other factors) not
accounted for in the predictors. Hence, the prediction of interannual anomalies remains a major unresolved research topic.
The predictions for ecosystem fluxes across FLUXNET by different ML techniques and by different explanatory variable
sets (RS vs RS+METEO) were highly consistent, indicating that the extracted patterns by the trained models were robust,
realistic and not subject to severe overfitting. The differences in predictions among the RS and RS+METEO setups were
slightly larger than among different ML methods, suggesting that future activities should concentrate on identifying new
driver variables to further improve the performance of fluxes predictions. Nevertheless, we recommend using the ensemble
median estimate of multiple ML techniques for analyzing global flux products because extrapolation beyond the FLUXNET-
sampled conditions may generate larger differences among methods than discernible from our cross-validation comparison.
The ML based models presented and extensively evaluated here form the basis of an extensive archive of global gridded flux
products, which is currently under development. The thorough cross-validation experiment presented in this paper helps
users understanding the products' strengths and weaknesses. The good performance of the ML methods, the availability of
an ensemble of them, and the detailed analysis of their uncertainties will make this archive an unprecedented data stream to
study the global land-atmosphere exchange of $CO_2$, water and energy.
**Appendix A: Median performance of the methods.**
In table A1 we reported, for both setups, the median value of skill metrics (MEF, RMSE, and absolute value of BIAS)
realized by singular ML and their related variability such estimated as the median absolute deviation (MAD) from the
median multiplied per 1.4826 (see Jung et al., 2009 for details)
**Appendix B: scatterplot between the observations and the predictions by the median ensemble of ML.**
In Fig. B1 and B2 we reported the scatterplots between eddy covariance observations and the modeled median ensemble
estimates respectively for RS and RS+METEO setup. We reported the overall eight days time series, and the comparison for
the across site variability, the mean seasonal cycle and the eight days deviations (or anomalies) from the mean seasonal
cycle.
**Appendix C Median value of site-by-site performance per vegetation and climate type**.
At follow we reported the median estimate of site-by-site $R^2$, RMSE, and absolute bias per PFT and climate zones.
**Acknowledgments**
G. Tramontana was supported by the GEOCARBON EU FP7 project (GA 283080). D. Papale, M. Jung and M. Reichstein
thank the support of the BACI H2020 (GA 640176) EU project. G. Camps-Valls wants to acknowledge the support by an
ERC Consolidator Grant with grant agreement 647423 (SEDAL). K. Ichii was supported by Environment Research and

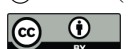


Technology Development Funds (2-1401) from the Ministry of the Environment of Japan and the JAXA Global Change Observation Mission (GCOM) project (#115). C. R. Schwalm was supported by National Aeronautics and Space Administration (NASA) Grants #NNX12AP74G, #NNX10AG01A, and #NNX11AO08A. M. A. Arain thanks the support of Natural Sciences and Engineering Research Council (NSREC) of Canada. P. Serrano Ortiz was partially supported by the Spanish Ministry of Economy and Competitiveness though the project CGL2014-52838-C2-R(GEISpain). S. Wolf acknowledges support from a Marie Curie International Outgoing Fellowship (European Commission, grant 300083). This work used Eddy Covariance data acquired by the FLUXNET community and in particular by the following networks: AmeriFlux (U.S. Department of Energy, Biological and Environmental Research, Terrestrial Carbon Program (DE-FG02-04ER63917 and DE-FG02-04ER63911)), AfriFlux, AsiaFlux, CarboAfrica, CarboEuropeIP, CarboItaly, CarboMont,ChinaFlux, Fluxnet-Canada (supported by CFCAS, NSERC, BIOCAP, Environment Canada, and NRCan), GreenGrass, KoFlux, LBA, NECC, OzFlux, TCOS-Siberia, USCCC. We acknowledge the financial support to the eddy covariance data harmonization provided by CarboEuropeIP, FAO-GTOS-TCO, iLEAPS, Max Planck Institute for Biogeochemistry, National Science Foundation, University of Tuscia and US Department of Energy and the databasing and technical support from Berkeley Water Center, Lawrence Berkeley National Laboratory, Microsoft Research eScience, Oak Ridge National Laboratory, University of California - Berkeley, University of Virginia.

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





**Table 1.** Distribution of flux tower sites across plant functional types (PFT) and climate zones.

| PFT | N° of sites | Climate zone | N° of sites |
|---|---|---|---|
| Evergreen needleleaf forest | 66 | Temperate | 111 |
| Grassland | 38 | Subtropical - Mediterranean | 47 |
| Cropland | 27 | Boreal | 34 |
| Deciduous broadleaf forest | 24 | Tropical | 14 |
| Evergreen broadleaf forest | 19 | Dry | 13 |
| Wetland | 17 | Artic | 5 |
| Shrubland | 12 | | |
| Mixed forest | 11 | | |
| Savannah | 10 | | |





**Table 2.** Selected predictors for both setup for $CO_2$ fluxes (GPP, TER and NEE) and energy fluxes (H, LE and Rn). List of acronyms:
Enhanced Vegetation Index (EVI), fraction of photosynthetically active radiation absorbed by a canopy (fPAR), Leaf Area Index (LAI),
daytime Land Surface Temperature ($LST_{Day}$) and nighttime Land Surface Temperature ($LST_{Night}$), Middle Infrared Reflectance (band 7)
($MIR^{(1)}$), Normalized Difference Vegetation Index (NDVI), Normalized Difference Water Index (NDWI), Plant Functional Type (PFT),
incoming global Radiation (Rg), top of atmosphere potential Radiation (Rpot), Index of Water Availability (IWA), Relative humidity (Rh),
Water Availability Index lower ($WAI_L$), and upper ($WAI_U$) (for details see supplementary material, Sect. S3), Mean Seasonal Cycle
(MSC). Interaction between A and B is shown as (A, B)

| Setup | Type of variability | $CO_2$ fluxes | Energy fluxes |
|---|---|---|---|
| RS | Spatial | PFT | PFT |
| | | Amplitude of MSC of EVI | Maximum of MSC of (fPAR, Rg) |
| | | Amplitude of MSC of $MIR^{(1)}$ | Minimum of MSC of (NDVI, Rg) |
| | | Maximum of MSC of $LST_{Day}$ | |
| | Spatial & seasonal | MSC LAI | MSC of (EVI, LST) |
| | | | Rpot |
| | Spatial, seasonal & interannual | NDWI | Rg |
| | | $LST_{Day}$ | $LST_{Day}$ |
| | | $LST_{Night}$ | Anomalies of $LST_{Night}$ |
| | | (NDVI, Rg) | Anomalies of (EVI, LST) |
| RS+METEO | Spatial | PFT | PFT |
| | | Amplitude of MSC of NDVI | Maximum of MSC of (NDVI, $WAI_U$) |
| | | Amplitude of MSC of band 4 BRDF reflectance$^{(2)}$ | Mean of MSC of band 6 BRDF reflectance$^{(2)}$ |
| | | Minimum of MSC of NDWI | |
| | | Amplitude of MSC of (NDVI, $WAI_L$) | |
| | Spatial & Seasonal | MSC of $LST_{Night}$ | Rpot |
| | | MSC of (FPAR, LST) | MSC of NDWI |
| | | MSC of (NDVI, Rg) | MSC of $LST_{Night}$ |
| | | | MSC of (NDVI, Rg, IWA) |
| | Spatial & Seasonal & Interannual | Tair | Rain |
| | | | Rg |
| | | | Rh |

$^{(1)}$ derived from the MOD13 product; $^{(2)}$ derived from MCD43 product.





**Table** 3. Statistics of the accuracy of predictions of $CO_2$ and energy fluxes made by the ensemble median estimate based on RS and
RS+METEO. For RMSE and BIAS, the reference units were $gCm^{-2}d^{-1}$ and $MJm^{-2}d^{-1}$, in the case of $CO_2$ fluxes (GPP, TER and NEE) and
energy fluxes (H, LE and Rn) respectively.

| Flux | RS | | | | | RS+METEO | | | | |
|---|---|---|---|---|---|---|---|---|---|---|
| | MEF | RMSE | P | ROV | BIAS | MEF | RMSE | $\rho$ | ROV | BIAS |
| $GPP_R$ | 0.71 | 1.56 | 0.85 | 0.69 | -0.02 | 0.70 | 1.59 | 0.84 | 0.73 | 0.09 |
| $GPP_L$ | 0.71 | 1.53 | 0.84 | 0.68 | -0.02 | 0.71 | 1.54 | 0.84 | 0.74 | 0.09 |
| $TER_R$ | 0.64 | 1.14 | 0.80 | 0.61 | -0.01 | 0.64 | 1.15 | 0.80 | 0.69 | 0.09 |
| $TER_L$ | 0.60 | 1.18 | 0.77 | 0.56 | -0.01 | 0.63 | 1.14 | 0.79 | 0.66 | 0.08 |
| NEE | 0.46 | 1.24 | 0.68 | 0.39 | 0.04 | 0.43 | 1.28 | 0.65 | 0.40 | -0.02 |
| H | 0.79 | 1.36 | 0.89 | 0.71 | -0.02 | 0.79 | 1.37 | 0.89 | 0.75 | 0.02 |
| LE | 0.76 | 1.37 | 0.87 | 0.71 | -0.07 | 0.75 | 1.39 | 0.87 | 0.73 | -0.01 |
| Rn | 0.92 | 1.51 | 0.96 | 0.90 | -0.01 | 0.91 | 1.55 | 0.96 | 0.93 | 0.08 |




**Table 4**: Mean values of the determination coefficient ($R^2$) by the pair-wise comparison of the models output and their residuals. We
compared different ML and same drivers (RS and RS+METEO respectively) or the same ML and different drivers (RS vs
RS+METEO). Numbers in brackets were the standard deviation of $R^2$. All correlations were statistically significant ($p < 0.001$).

| Fluxes | Correlation among models output | | | Correlation among models residuals | | |
|---|---|---|---|---|---|---|
| | RS | RS+METEO | RS vs RS+METEO | RS | RS+METEO | RS vs RS+METEO |
| $GPP_R$ | 0.95 (0.02) | 0.95 (0.02) | 0.89 (0.02) | 0.88 (0.04) | 0.87 (0.04) | 0.74 (0.04) |
| $GPP_L$ | 0.95 (0.02) | 0.94 (0.02) | 0.88 (0.02) | 0.88 (0.04) | 0.86 (0.04) | 0.72 (0.04) |
| $TER_R$ | 0.91 (0.03) | 0.94 (0.03) | 0.86 (0.04) | 0.86 (0.05) | 0.88 (0.05) | 0.75 (0.06) |
| $TER_L$ | 0.92 (0.03) | 0.93 (0.03) | 0.85 (0.03) | 0.89 (0.04) | 0.88 (0.05) | 0.77 (0.05) |
| NEE | 0.84 (0.06) | 0.84 (0.07) | 0.75 (0.08) | 0.88 (0.05) | 0.87 (0.06) | 0.80 (0.06) |
| H | 0.94 (0.02) | 0.96 (0.02) | 0.93 (0.03) | 0.80 (0.06) | 0.87 (0.05) | 0.76 (0.08) |
| LE | 0.94 (0.02) | 0.96 (0.01) | 0.90 (0.02) | 0.83 (0.05) | 0.88 (0.04) | 0.73 (0.04) |
| Rn | 0.98 (0.01) | 0.99 (0.00) | 0.97 (0.01) | 0.79 (0.08) | 0.86 (0.03) | 0.71 (0.12) |




**Table 5**: $R^2$ and RMSE for the comparison among sites, mean seasonal cycle and anomalies. The last two columns showed the consistency
between the median estimates of the two setups. For RMSE, the reference units were $gCm^{-2}d^{-1}$ and $MJm^{-2}d^{-1}$, in the case of $CO_2$ fluxes
(GPP, TER and NEE) and energy fluxes (H, LE and Rn) respectively.

| Fluxes | RS *vs.* OBS | | RS+METEO *vs.* OBS | | RS *vs.* RS+METEO | |
|---|---|---|---|---|---|---|
| | $R^2$ | RMSE | $R^2$ | RMSE | $R^2$ | RMSE |
| Across-sites | | | | | | |
| $GPP_R$ | 0.78 | 0.80 | 0.77 | 0.82 | 0.95 | 0.34 |
| $GPP_L$ | 0.78 | 0.77 | 0.79 | 0.75 | 0.94 | 0.36 |
| $TER_R$ | 0.68 | 0.73 | 0.61 | 0.81 | 0.92 | 0.32 |
| $TER_L$ | 0.72 | 0.60 | 0.71 | 0.61 | 0.92 | 0.27 |
| NEE | 0.48 | 0.61 | 0.46 | 0.61 | 0.83 | 0.22 |
| H | 0.81 | 0.68 | 0.81 | 0.68 | 0.97 | 0.25 |
| LE | 0.79 | 0.74 | 0.75 | 0.80 | 0.93 | 0.33 |
| Rn | 0.80 | 0.93 | 0.79 | 0.96 | 0.96 | 0.38 |
| Mean Seasonal Cycle | | | | | | |
| $GPP_R$ | 0.76 | 1.03 | 0.77 | 1.02 | 0.93 | 0.48 |
| $GPP_L$ | 0.77 | 1.00 | 0.77 | 0.99 | 0.93 | 0.50 |
| $TER_R$ | 0.71 | 0.62 | 0.71 | 0.62 | 0.92 | 0.29 |
| $TER_L$ | 0.67 | 0.64 | 0.68 | 0.63 | 0.92 | 0.29 |
| NEE | 0.61 | 0.83 | 0.59 | 0.84 | 0.93 | 0.24 |
| H | 0.86 | 0.89 | 0.86 | 0.87 | 0.97 | 0.36 |
| LE | 0.87 | 0.79 | 0.87 | 0.79 | 0.95 | 0.45 |
| Rn | 0.98 | 0.74 | 0.98 | 0.74 | 0.99 | 0.43 |
| Anomalies | | | | | | |
| $GPP_R$ | 0.18 | 0.67 | 0.12 | 0.68 | 0.38 | 0.32 |
| $GPP_L$ | 0.16 | 0.67 | 0.11 | 0.68 | 0.37 | 0.31 |
| $TER_R$ | 0.14 | 0.48 | 0.15 | 0.48 | 0.36 | 0.17 |
| $TER_L$ | 0.10 | 0.58 | 0.13 | 0.57 | 0.35 | 0.18 |
| NEE | 0.13 | 0.56 | 0.13 | 0.55 | 0.43 | 0.20 |
| H | 0.43 | 0.81 | 0.41 | 0.81 | 0.77 | 0.34 |
| LE | 0.21 | 0.78 | 0.21 | 0.77 | 0.46 | 0.32 |
| Rn | 0.57 | 0.81 | 0.54 | 0.83 | 0.84 | 0.41 |




**Table A1**: Accuracy of $CO_2$ and energy fluxes predicted by machine learning methods based on RS and RS+METEO dataset. The median
value of methods and the variability (in brackets) estimated as the median absolute deviation (MAD) from the median multiplied per
1.4826 (as reported in Jung et al., 2009) were reported.

| FLUXES | RS | | | RS+METEO | | |
|---|---|---|---|---|---|---|
| | MEF | RMSE | Abs BIAS | MEF | RMSE | Abs BIAS |
| GPP | 0.698 (±0.012) | 1.604 (±0.033) | 0.022 (±0.019) | 0.694 (±0.012) | 1.614 (±0.032) | 0.073 (±0.011) |
| $GPP_{HB}$ | 0.700 (±0.009) | 1.564 (±0.024) | 0.023 (±0.024) | 0.701 (±0.008) | 1.561 (±0.020) | 0.083 (±0.011) |
| TER | 0.612 (±0.022) | 1.183 (±0.033) | 0.026 (±0.025) | 0.623 (±0.005) | 1.166 (±0.008) | 0.089 (±0.033) |
| $TER_{HB}$ | 0.571 (±0.016) | 1.218 (±0.023) | 0.019 (±0.017) | 0.609 (±0.001) | 1.163 (±0.002) | 0.079 (±0.017) |
| NEE | 0.433 (±0.017) | 1.270 (±0.019) | 0.024 (±0.021) | 0.407 (±0.029) | 1.298 (±0.032) | 0.014 (±0.003) |
| H | 0.767 (±0.015) | 1.426 (±0.047) | 0.014 (±0.005) | 0.776 (±0.008) | 1.397 (±0.025) | 0.022 (±0.009) |
| LE | 0.739 (±0.015) | 1.418 (±0.042) | 0.052 (±0.046) | 0.734 (±0.003) | 1.434 (±0.009) | 0.023 (±0.008) |
| Rn | 0.909 (±0.009) | 1.589 (±0.082) | 0.030 (±0.025) | 0.908 (±0.008) | 1.600 (±0.070) | 0.073 (±0.015) |






**Table C1.** Median site-by-site $R^2$ of the $CO_2$ fluxes per PFT and climate zones. ENF was evergreen needle leaf forest, DBF deciduous broadleaf forest, EBF Evergreen broadleaf forest, MF mixed forest, SHR shrubland, SAV Savannah, GRA Grassland, CRO cropland, WET Wetland, Trop Tropical, SubTrop Subtropical, Dry Dryland, Tmp Temperate, TmpCont Temperate-continental, Bor Boreal, Cold cold environment or Iceland covered by ice.

| CAT | $GPP_R$ | | $GPP_L$ | | $TER_R$ | | $TER_L$ | | NEE | |
|---|---|---|---|---|---|---|---|---|---|---|
| | RS | RS+METEO | RS | RS+METEO | RS | RS+METEO | RS | RS+METEO | RS | RS+METEO |
| ENF | 0.87 (0.10) | 0.86 (0.10) | 0.85 (0.12) | 0.86 (0.12) | 0.81 (0.15) | 0.85 (0.11) | 0.75 (0.24) | 0.76 (0.20) | 0.50 (0.34) | 0.55 (0.30) |
| DBF | 0.89 (0.07) | 0.87 (0.09) | 0.87 (0.07) | 0.88 (0.08) | 0.81 (0.12) | 0.83 (0.13) | 0.76 (0.14) | 0.76 (0.14) | 0.72 (0.16) | 0.68 (0.17) |
| EBF | 0.50 (0.29) | 0.48 (0.20) | 0.48 (0.29) | 0.44 (0.28) | 0.34 (0.34) | 0.49 (0.35) | 0.15 (0.18) | 0.29 (0.20) | 0.26 (0.23) | 0.24 (0.26) |
| MF | 0.91 (0.06) | 0.95 (0.02) | 0.91 (0.03) | 0.95 (0.04) | 0.85 (0.10) | 0.90 (0.07) | 0.84 (0.10) | 0.86 (0.15) | 0.73 (0.10) | 0.75 (0.09) |
| SHR | 0.67 (0.30) | 0.71 (0.28) | 0.67 (0.36) | 0.72 (0.23) | 0.80 (0.13) | 0.78 (0.24) | 0.68 (0.18) | 0.66 (0.38) | 0.37 (0.38) | 0.41 (0.31) |
| SAV | 0.75 (0.13) | 0.70 (0.13) | 0.72 (0.05) | 0.67 (0.17) | 0.65 (0.07) | 0.72 (0.11) | 0.55 (0.16) | 0.61 (0.10) | 0.38 (0.20) | 0.34 (0.29) |
| GRA | 0.69 (0.27) | 0.62 (0.33) | 0.69 (0.25) | 0.60 (0.32) | 0.70 (0.25) | 0.73 (0.25) | 0.66 (0.20) | 0.72 (0.21) | 0.40 (0.29) | 0.36 (0.30) |
| CRO | 0.58 (0.41) | 0.44 (0.36) | 0.56 (0.41) | 0.45 (0.31) | 0.78 (0.17) | 0.76 (0.15) | 0.68 (0.22) | 0.65 (0.23) | 0.35 (0.46) | 0.33 (0.43) |
| WET | 0.87 (0.11) | 0.91 (0.07) | 0.85 (0.12) | 0.87 (0.09) | 0.78 (0.19) | 0.83 (0.14) | 0.65 (0.17) | 0.74 (0.20) | 0.64 (0.16) | 0.61 (0.24) |
| Trop | 0.32 (0.46) | 0.40 (0.39) | 0.63 (0.23) | 0.31 (0.32) | 0.25 (0.23) | 0.34 (0.47) | 0.11 (0.13) | 0.26 (0.14) | 0.28 (0.35) | 0.21 (0.30) |
| SubTrop | 0.64 (0.26) | 0.66 (0.28) | 0.65 (0.26) | 0.65 (0.24) | 0.64 (0.25) | 0.66 (0.26) | 0.52 (0.24) | 0.55 (0.28) | 0.39 (0.37) | 0.39 (0.26) |
| Dry | 0.47 (0.27) | 0.40 (0.33) | 0.50 (0.25) | 0.38 (0.30) | 0.62 (0.25) | 0.62 (0.38) | 0.55 (0.19) | 0.55 (0.39) | 0.21 (0.29) | 0.11 (0.14) |
| Tmp | 0.81 (0.19) | 0.74 (0.24) | 0.83 (0.14) | 0.78 (0.22) | 0.78 (0.13) | 0.77 (0.18) | 0.68 (0.20) | 0.72 (0.17) | 0.56 (0.28) | 0.47 (0.34) |
| TmpCont | 0.86 (0.09) | 0.82 (0.16) | 0.84 (0.11) | 0.80 (0.17) | 0.81 (0.12) | 0.78 (0.14) | 0.75 (0.17) | 0.76 (0.15) | 0.54 (0.42) | 0.53 (0.36) |
| Bor | 0.90 (0.07) | 0.90 (0.07) | 0.92 (0.06) | 0.89 (0.07) | 0.90 (0.05) | 0.91 (0.04) | 0.86 (0.08) | 0.89 (0.06) | 0.59 (0.31) | 0.59 (0.25) |
| Cold | 0.56 (0.57) | 0.50 (0.56) | 0.49 (0.62) | 0.46 (0.59) | 0.84 (0.20) | 0.86 (0.13) | 0.50 (0.38) | 0.55 (0.23) | 0.47 (0.56) | 0.45 (0.57) |






**Table C2.** Median site-by-site RMSE of the $CO_2$ fluxes per PFT and climate zones. ENF was svergreen needle leaf forest, DBF deciduous broadleaf forest, EBF Evergreen broadleaf forest, MF mixed forest, SHR shrubland, SAV Savannah, GRA Grassland, CRO cropland, WET Wetland, Trop Tropical, SubTrop Subtropical, Dry Dryland, Tmp Temperate, TmpCont Temperate-continental, Bor Boreal, Cold cold environment or Iceland covered by ice.

| CAT | GPP$_R$ (gCm⁻²d⁻¹) | | GPP$_L$ (gCm⁻²d⁻¹) | | TER$_R$ (gCm⁻²d⁻¹) | | TER$_L$ (gCm⁻²d⁻¹) | | NEE (gCm⁻²d⁻¹) | |
|---|---|---|---|---|---|---|---|---|---|---|
| | RS | RS+METEO | RS | RS+METEO | RS | RS+METEO | RS | RS+METEO | RS | RS+METEO |
| ENF | 1.05 (0.60) | 1.12 (0.60) | 1.04 (0.59) | 1.14 (0.66) | 0.82 (0.50) | 0.80 (0.52) | 0.87 (0.60) | 0.91 (0.68) | 0.87 (0.51) | 0.86 (0.53) |
| DBF | 1.21 (0.78) | 1.35 (0.59) | 1.17 (0.68) | 1.36 (0.62) | 0.68 (0.26) | 0.76 (0.33) | 0.76 (0.33) | 0.93 (0.44) | 1.28 (0.39) | 1.28 (0.39) |
| EBF | 1.70 (0.55) | 1.64 (0.85) | 1.65 (0.70) | 1.46 (0.51) | 1.23 (0.69) | 1.48 (0.85) | 1.88 (1.23) | 1.71 (0.73) | 1.15 (0.48) | 1.15 (0.45) |
| MF | 0.87 (0.17) | 0.76 (0.45) | 0.89 (0.27) | 0.97 (0.56) | 0.65 (0.18) | 0.73 (0.42) | 0.79 (0.14) | 0.79 (0.18) | 0.91 (0.47) | 0.81 (0.29) |
| SHR | 0.73 (0.47) | 0.78 (0.46) | 0.69 (0.44) | 0.77 (0.37) | 0.50 (0.33) | 0.70 (0.41) | 0.50 (0.34) | 0.55 (0.36) | 0.57 (0.31) | 0.52 (0.15) |
| SAV | 0.83 (0.44) | 0.81 (0.18) | 0.87 (0.45) | 0.84 (0.18) | 0.80 (0.53) | 0.68 (0.41) | 0.86 (0.55) | 0.77 (0.38) | 0.71 (0.36) | 0.69 (0.31) |
| GRA | 1.22 (0.64) | 1.22 (0.60) | 1.18 (0.68) | 1.20 (0.62) | 1.00 (0.48) | 1.01 (0.54) | 0.99 (0.58) | 0.95 (0.52) | 0.76 (0.61) | 0.85 (0.49) |
| CRO | 1.69 (1.38) | 2.30 (1.02) | 1.57 (1.42) | 2.24 (1.10) | 0.87 (0.46) | 0.90 (0.57) | 0.80 (0.51) | 0.98 (0.57) | 1.42 (0.90) | 1.44 (0.70) |
| WET | 1.04 (0.95) | 0.93 (0.77) | 1.03 (0.96) | 0.78 (0.53) | 1.04 (0.87) | 0.98 (0.82) | 1.07 (0.51) | 1.02 (0.51) | 0.46 (0.19) | 0.64 (0.26) |
| Trop | 1.93 (0.46) | 1.74 (1.01) | 2.24 (0.62) | 1.56 (0.78) | 2.07 (0.69) | 1.55 (0.87) | 2.47 (0.74) | 2.05 (0.43) | 1.28 (0.29) | 1.17 (0.46) |
| SubTrop | 1.37 (0.55) | 1.40 (0.61) | 1.37 (0.56) | 1.38 (0.57) | 1.03 (0.46) | 1.00 (0.41) | 1.08 (0.36) | 1.11 (0.40) | 1.13 (0.63) | 1.15 (0.62) |
| Dry | 0.60 (0.24) | 0.78 (0.36) | 0.63 (0.16) | 0.74 (0.30) | 0.49 (0.10) | 0.54 (0.20) | 0.58 (0.26) | 0.67 (0.32) | 0.41 (0.13) | 0.46 (0.15) |
| Tmp | 1.73 (1.02) | 1.82 (0.99) | 1.73 (0.98) | 1.71 (1.03) | 1.09 (0.54) | 1.17 (0.67) | 1.24 (0.57) | 1.31 (0.59) | 1.43 (0.59) | 1.40 (0.61) |
| TmpCont | 1.01 (0.42) | 1.29 (0.59) | 1.00 (0.45) | 1.26 (0.57) | 0.71 (0.30) | 0.75 (0.38) | 0.74 (0.31) | 0.79 (0.34) | 0.95 (0.39) | 1.02 (0.43) |
| Bor | 0.66 (0.27) | 0.70 (0.36) | 0.66 (0.27) | 0.67 (0.33) | 0.48 (0.27) | 0.47 (0.27) | 0.48 (0.16) | 0.45 (0.21) | 0.50 (0.32) | 0.48 (0.22) |
| Cold | 0.44 (0.04) | 0.58 (0.42) | 0.51 (0.24) | 0.46 (0.32) | 0.41 (0.06) | 0.23 (0.06) | 0.57 (0.16) | 0.29 (0.12) | 0.51 (0.21) | 0.54 (0.35) |





**Table C3.** Median site-by-site absolute bias of the CO$_2$ fluxes per PFT and climate zones. ENF was evergreen needle leaf forest, DBF deciduous broadleaf forest, EBF Evergreen broadleaf forest, MF mixed forest, SHR shrubland, SAV Savannah, GRA Grassland, CRO cropland, WET Wetland, Trop Tropical, SubTrop Subtropical, Dry Dryland, Tmp Temperate, TmpCont Temperate-continental, Bor Boreal, Cold cold environment or Iceland covered by ice.

| CAT | GPP$_R$ (gCm$^{-2}$d$^{-1}$) | | GPP$_L$ (gCm$^{-2}$d$^{-1}$) | | TER$_R$ (gCm$^{-2}$d$^{-1}$) | | TER$_L$ (gCm$^{-2}$d$^{-1}$) | | NEE (gCm$^{-2}$d$^{-1}$) | |
|---|---|---|---|---|---|---|---|---|---|---|
| | RS | RS+METEO | RS | RS+METEO | RS | RS+METEO | RS | RS+METEO | RS | RS+METEO |
| ENF | 0.53 (0.46) | 0.54 (0.56) | 0.45 (0.42) | 0.48 (0.50) | 0.47 (0.47) | 0.50 (0.54) | 0.42 (0.40) | 0.41 (0.43) | 0.39 (0.44) | 0.32 (0.36) |
| DBF | 0.43 (0.38) | 0.56 (0.59) | 0.42 (0.36) | 0.50 (0.52) | 0.29 (0.32) | 0.35 (0.35) | 0.39 (0.33) | 0.42 (0.34) | 0.60 (0.28) | 0.55 (0.30) |
| EBF | 0.82 (0.91) | 0.77 (0.50) | 0.75 (0.81) | 0.76 (0.48) | 0.88 (0.98) | 0.84 (0.72) | 0.76 (0.81) | 0.93 (0.65) | 0.36 (0.45) | 0.46 (0.44) |
| MF | 0.47 (0.20) | 0.34 (0.38) | 0.38 (0.29) | 0.57 (0.29) | 0.39 (0.28) | 0.41 (0.13) | 0.37 (0.15) | 0.30 (0.35) | 0.34 (0.49) | 0.32 (0.36) |
| SHR | 0.38 (0.37) | 0.54 (0.49) | 0.38 (0.44) | 0.39 (0.47) | 0.36 (0.38) | 0.50 (0.43) | 0.31 (0.40) | 0.32 (0.23) | 0.27 (0.27) | 0.28 (0.24) |
| SAV | 0.42 (0.40) | 0.36 (0.21) | 0.35 (0.40) | 0.23 (0.15) | 0.43 (0.41) | 0.35 (0.23) | 0.42 (0.37) | 0.31 (0.10) | 0.23 (0.21) | 0.19 (0.10) |
| GRA | 0.60 (0.59) | 0.48 (0.49) | 0.60 (0.56) | 0.52 (0.55) | 0.38 (0.29) | 0.36 (0.37) | 0.44 (0.39) | 0.38 (0.38) | 0.17 (0.20) | 0.31 (0.31) |
| CRO | 0.47 (0.37) | 0.66 (0.44) | 0.36 (0.33) | 0.56 (0.47) | 0.29 (0.32) | 0.25 (0.22) | 0.29 (0.32) | 0.30 (0.29) | 0.41 (0.31) | 0.56 (0.55) |
| WET | 0.54 (0.64) | 0.28 (0.41) | 0.55 (0.62) | 0.29 (0.25) | 0.72 (0.35) | 0.48 (0.52) | 0.69 (0.29) | 0.50 (0.51) | 0.24 (0.19) | 0.30 (0.25) |
| Trop | 1.66 (1.31) | 0.67 (0.79) | 1.71 (1.23) | 0.77 (0.86) | 1.73 (0.88) | 1.16 (1.19) | 1.94 (0.81) | 1.21 (0.67) | 0.52 (0.57) | 0.38 (0.55) |
| SubTrop | 0.54 (0.45) | 0.55 (0.43) | 0.50 (0.38) | 0.52 (0.55) | 0.46 (0.44) | 0.53 (0.47) | 0.47 (0.35) | 0.42 (0.37) | 0.34 (0.44) | 0.37 (0.34) |
| Dry | 0.31 (0.20) | 0.33 (0.26) | 0.33 (0.38) | 0.36 (0.29) | 0.24 (0.21) | 0.32 (0.35) | 0.34 (0.21) | 0.43 (0.26) | 0.14 (0.08) | 0.22 (0.14) |
| Tmp | 0.72 (0.55) | 0.77 (0.71) | 0.66 (0.59) | 0.63 (0.56) | 0.50 (0.46) | 0.47 (0.50) | 0.51 (0.55) | 0.41 (0.45) | 0.46 (0.43) | 0.51 (0.41) |
| TmpCont | 0.45 (0.35) | 0.60 (0.52) | 0.39 (0.35) | 0.57 (0.47) | 0.37 (0.28) | 0.29 (0.25) | 0.37 (0.33) | 0.38 (0.37) | 0.35 (0.40) | 0.55 (0.55) |
| Bor | 0.36 (0.30) | 0.32 (0.34) | 0.32 (0.24) | 0.27 (0.31) | 0.32 (0.40) | 0.32 (0.33) | 0.31 (0.35) | 0.26 (0.32) | 0.27 (0.26) | 0.23 (0.26) |
| Cold | 0.07 (0.00) | 0.08 (0.09) | 0.08 (0.12) | 0.15 (0.06) | 0.34 (0.04) | 0.12 (0.06) | 0.34 (0.06) | 0.15 (0.01) | 0.37 (0.15) | 0.27 (0.27) |



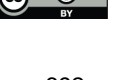

**Table C4.** Median site-by-site $R^2$ of the energy fluxes per PFT and climate zones. ENF was evergreen needle leaf forest, DBF deciduous
broadleaf forest, EBF Evergreen broadleaf forest, MF mixed forest, SHR shrubland, SAV Savannah, GRA Grassland, CRO cropland,
WET Wetland, Trop Tropical, SubTrop Subtropical, Dry Dryland, Tmp Temperate, TmpCont Temperate-continental, Bor Boreal, Cold
cold environment or Iceland covered by ice.

| CAT | H | | LE | | Rn | |
| --- | --- | --- | --- | --- | --- | --- |
| | RS | RS+METEO | RS | RS+METEO | RS | RS+METEO |
| ENF | 0.87 (0.10) | 0.86 (0.10) | 0.83 (0.10) | 0.84 (0.11) | 0.97 (0.02) | 0.97 (0.02) |
| DBF | 0.76 (0.18) | 0.74 (0.12) | 0.87 (0.05) | 0.87 (0.07) | 0.97 (0.01) | 0.97 (0.02) |
| EBF | 0.85 (0.13) | 0.82 (0.17) | 0.56 (0.30) | 0.52 (0.42) | 0.95 (0.05) | 0.96 (0.03) |
| MF | 0.85 (0.06) | 0.82 (0.10) | 0.91 (0.07) | 0.89 (0.06) | 0.97 (0.02) | 0.96 (0.02) |
| SHR | 0.83 (0.15) | 0.83 (0.17) | 0.73 (0.29) | 0.77 (0.23) | 0.98 (0.01) | 0.97 (0.01) |
| SAV | 0.74 (0.25) | 0.77 (0.26) | 0.85 (0.06) | 0.78 (0.11) | 0.86 (0.05) | 0.88 (0.10) |
| GRA | 0.72 (0.22) | 0.71 (0.22) | 0.85 (0.11) | 0.83 (0.16) | 0.96 (0.02) | 0.96 (0.02) |
| CRO | 0.70 (0.16) | 0.66 (0.18) | 0.79 (0.14) | 0.80 (0.14) | 0.97 (0.02) | 0.96 (0.02) |
| WET | 0.81 (0.06) | 0.78 (0.14) | 0.86 (0.10) | 0.84 (0.06) | 0.94 (0.02) | 0.92 (0.06) |
| Trop | 0.52 (0.18) | 0.60 (0.32) | 0.56 (0.38) | 0.50 (0.44) | 0.86 (0.14) | 0.89 (0.13) |
| SubTrop | 0.81 (0.18) | 0.82 (0.18) | 0.78 (0.13) | 0.80 (0.13) | 0.96 (0.03) | 0.96 (0.02) |
| Dry | 0.87 (0.07) | 0.86 (0.13) | 0.80 (0.07) | 0.79 (0.14) | 0.90 (0.06) | 0.93 (0.05) |
| Tmp | 0.78 (0.14) | 0.78 (0.13) | 0.86 (0.11) | 0.83 (0.13) | 0.97 (0.02) | 0.96 (0.02) |
| TmpCont | 0.72 (0.18) | 0.69 (0.18) | 0.83 (0.08) | 0.84 (0.09) | 0.97 (0.02) | 0.96 (0.02) |
| Bor | 0.90 (0.07) | 0.89 (0.08) | 0.92 (0.05) | 0.92 (0.03) | 0.98 (0.01) | 0.97 (0.02) |
| Cold | 0.83 (0.12) | 0.57 (0.19) | 0.83 (0.08) | 0.82 (0.07) | 0.94 (0.03) | 0.85 (0.13) |



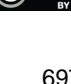

**Table C5.** Median site-by-site RMSE of the energy fluxes per PFT and climate zones. ENF was evergreen needle leaf forest, DBF
deciduous broadleaf forest, EBF Evergreen broadleaf forest, MF mixed forest, SHR shrubland, SAV Savannah, GRA Grassland, CRO
cropland, WET Wetland, Trop Tropical, SubTrop Subtropical, Dry Dryland, Tmp Temperate, TmpCont Temperate-continental, Bor
Boreal, Cold cold environment or Iceland covered by ice.

| CAT | H ($MJm^{-2}d^{-1}$) | | LE ($MJm^{-2}d^{-1}$) | | Rn ($MJm^{-2}d^{-1}$) | |
|---|---|---|---|---|---|---|
| | RS | RS+METEO | RS | RS+METEO | RS | RS+METEO |
| ENF | 1.09 (0.25) | 1.16 (0.25) | 1.00 (0.56) | 1.02 (0.55) | 1.27 (0.68) | 1.26 (0.57) |
| DBF | 1.30 (0.43) | 1.31 (0.38) | 1.22 (0.26) | 1.14 (0.46) | 1.11 (0.42) | 1.24 (0.41) |
| EBF | 1.14 (0.60) | 1.29 (0.76) | 1.55 (0.39) | 1.60 (0.46) | 1.33 (0.43) | 1.14 (0.56) |
| MF | 1.18 (0.44) | 1.12 (0.42) | 0.82 (0.37) | 1.15 (0.54) | 1.14 (0.45) | 1.09 (0.43) |
| SHR | 1.21 (0.46) | 1.14 (0.28) | 1.12 (0.41) | 1.11 (0.56) | 1.37 (0.80) | 1.01 (0.43) |
| SAV | 1.23 (0.25) | 1.20 (0.22) | 1.32 (0.56) | 1.35 (0.30) | 1.10 (0.33) | 1.19 (0.60) |
| GRA | 1.14 (0.35) | 1.08 (0.47) | 1.09 (0.34) | 1.32 (0.54) | 1.48 (0.83) | 1.48 (0.90) |
| CRO | 1.24 (0.45) | 1.36 (0.33) | 1.51 (0.61) | 1.54 (0.35) | 1.24 (0.52) | 1.23 (0.26) |
| WET | 0.97 (0.36) | 1.22 (0.60) | 0.88 (0.13) | 0.90 (0.18) | 1.42 (0.51) | 1.65 (0.71) |
| Trop | 0.98 (0.51) | 1.19 (0.63) | 1.60 (0.52) | 1.62 (0.41) | 1.33 (0.73) | 1.03 (0.48) |
| SubTrop | 1.28 (0.38) | 1.32 (0.46) | 1.36 (0.62) | 1.36 (0.53) | 1.40 (0.40) | 1.33 (0.49) |
| Dry | 1.07 (0.24) | 1.05 (0.50) | 1.21 (0.33) | 1.27 (0.52) | 1.61 (0.75) | 2.02 (0.93) |
| Tmp | 1.18 (0.23) | 1.15 (0.33) | 1.18 (0.43) | 1.17 (0.49) | 1.10 (0.36) | 1.14 (0.47) |
| TmpCont | 1.30 (0.42) | 1.35 (0.37) | 1.25 (0.41) | 1.47 (0.37) | 1.17 (0.65) | 1.16 (0.54) |
| Bor | 0.98 (0.23) | 1.05 (0.26) | 0.70 (0.26) | 0.61 (0.20) | 0.88 (0.31) | 1.08 (0.50) |
| Cold | 1.03 (0.36) | 1.50 (0.55) | 1.00 (0.23) | 1.03 (0.45) | 1.47 (0.18) | 2.04 (0.19) |





**Table C6.** Median site-by-site absolute bias for energy fluxes. . ENF was evergreen needle leaf forest, DBF deciduous broadleaf forest,
EBF Evergreen broadleaf forest, MF mixed forest, SHR shrubland, SAV Savannah, GRA Grassland, CRO cropland, WET Wetland, Trop
Tropical, SubTrop Subtropical, Dry Dryland, Tmp Temperate, TmpCont Temperate-continental, Bor Boreal, Cold cold environment or
Iceland covered by ice.

| CAT | H (MJm⁻²d⁻¹) | | LE (MJm⁻²d⁻¹) | | Rn (MJm⁻²d⁻¹) | |
|---|---|---|---|---|---|---|
| | RS | RS+METEO | RS | RS+METEO | RS | RS+METEO |
| ENF | 0.44 (0.40) | 0.40 (0.33) | 0.42 (0.41) | 0.44 (0.49) | 0.78 (0.63) | 0.64 (0.61) |
| DBF | 0.60 (0.35) | 0.66 (0.35) | 0.57 (0.56) | 0.49 (0.50) | 0.38 (0.28) | 0.61 (0.49) |
| EBF | 0.38 (0.48) | 0.55 (0.46) | 0.97 (0.79) | 0.88 (0.70) | 0.88 (0.51) | 0.62 (0.43) |
| MF | 0.48 (0.40) | 0.26 (0.31) | 0.34 (0.40) | 0.64 (0.52) | 0.56 (0.45) | 0.56 (0.57) |
| SHR | 0.34 (0.43) | 0.47 (0.52) | 0.41 (0.41) | 0.50 (0.43) | 0.62 (0.76) | 0.44 (0.52) |
| SAV | 0.68 (0.35) | 0.56 (0.15) | 0.63 (0.80) | 0.40 (0.15) | 0.27 (0.22) | 0.63 (0.55) |
| GRA | 0.51 (0.39) | 0.40 (0.24) | 0.38 (0.38) | 0.57 (0.50) | 0.97 (0.81) | 0.81 (1.03) |
| CRO | 0.23 (0.21) | 0.24 (0.24) | 0.36 (0.38) | 0.41 (0.50) | 0.66 (0.58) | 0.68 (0.39) |
| WET | 0.47 (0.51) | 0.67 (0.37) | 0.54 (0.41) | 0.38 (0.21) | 0.34 (0.34) | 0.83 (0.78) |
| Trop | 0.37 (0.51) | 0.67 (0.47) | 0.97 (0.79) | 1.24 (0.82) | 0.94 (1.10) | 0.63 (0.60) |
| SubTrop | 0.58 (0.59) | 0.50 (0.39) | 0.62 (0.58) | 0.58 (0.56) | 0.83 (0.71) | 0.70 (0.55) |
| Dry | 0.68 (0.62) | 0.55 (0.56) | 0.21 (0.14) | 0.30 (0.26) | 1.06 (0.55) | 1.61 (0.91) |
| Tmp | 0.38 (0.23) | 0.34 (0.31) | 0.49 (0.46) | 0.56 (0.54) | 0.65 (0.49) | 0.68 (0.58) |
| TmpCont | 0.49 (0.41) | 0.40 (0.46) | 0.44 (0.51) | 0.53 (0.50) | 0.69 (0.72) | 0.61 (0.58) |
| Bor | 0.33 (0.32) | 0.38 (0.24) | 0.22 (0.16) | 0.23 (0.24) | 0.38 (0.27) | 0.50 (0.47) |
| Cold | 0.43 (0.46) | 0.71 (0.11) | 0.56 (0.31) | 0.39 (0.18) | 0.30 (0.29) | 0.86 (0.58) |





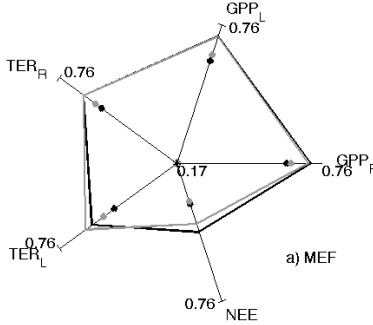
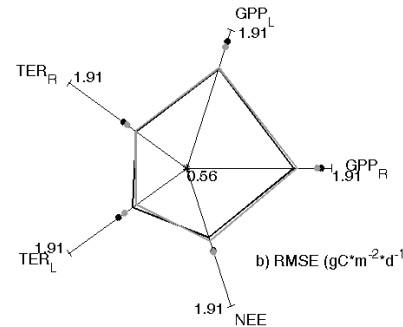

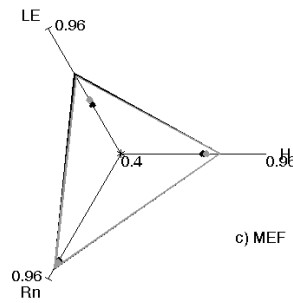
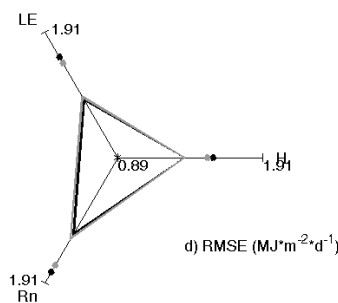

**Fig. 1**: Spider plot of MEF (first column) and RMSE (second column) for $CO_2$ (first row) and energy fluxes (second row) showing the consistency of prediction made by RS (Black line) and RS+METEO (grey lines) setups. The lines were the ensemble median estimate of ML; we also showed the performance of multiple regressions trained with RS (black point) and RS+METEO (gray points). $GPP_R$ and $GPP_L$ were respectively gross primary production estimated following Reichstein et al. (2005) and Lasslop et al. (2010), $TER_R$ and $TER_L$ the total ecosystem respiration estimated following Reichstein et al. (2005) and Lasslop et al. (2010), NEE net ecosystem exchange, H the sensible heat, LE the latent heat and Rn the net radiation.





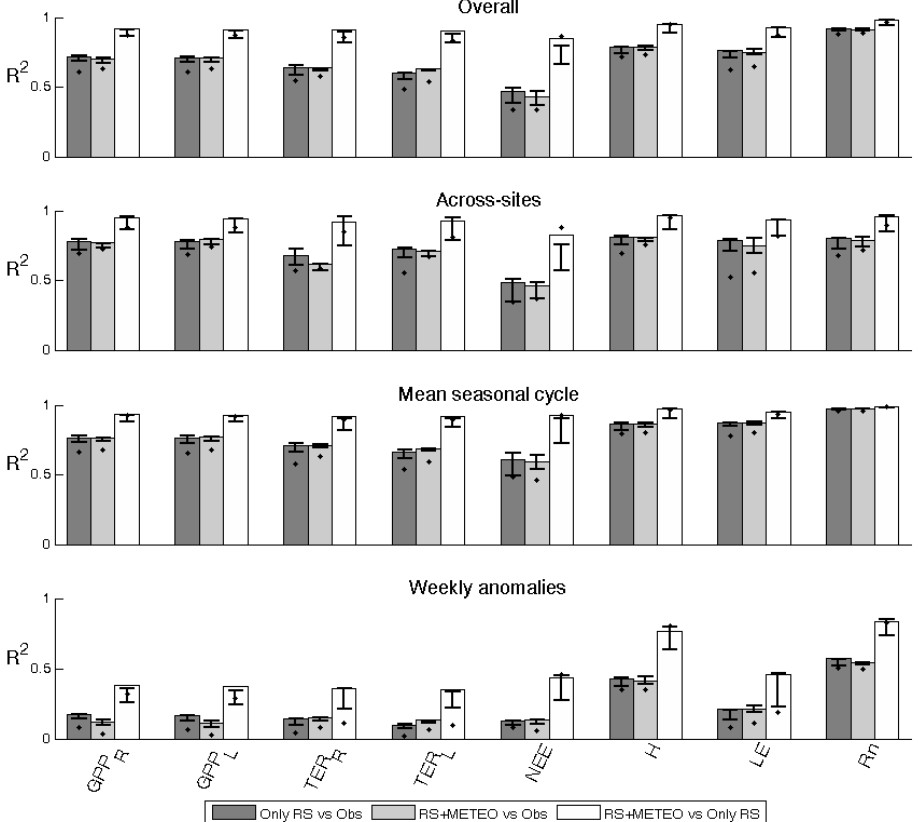

**Fig. 2:** Coefficients of determination ($R^2$) from the comparison of overall time series, across-sites, mean seasonal cycle, and the anomalies, in particular: the determination coefficients between predictions by the ensemble median estimate of RS setup and observation (dark grey bars), between predictions by the ensemble median estimate of RS+METEO setup and observation (light grey bars), and between the two ensembles median estimate (white bars). Whiskers were the higher and lower $R^2$ when the comparisons were made among the singular ML. The comparison of output by the multiple regressions was also shown (black points).



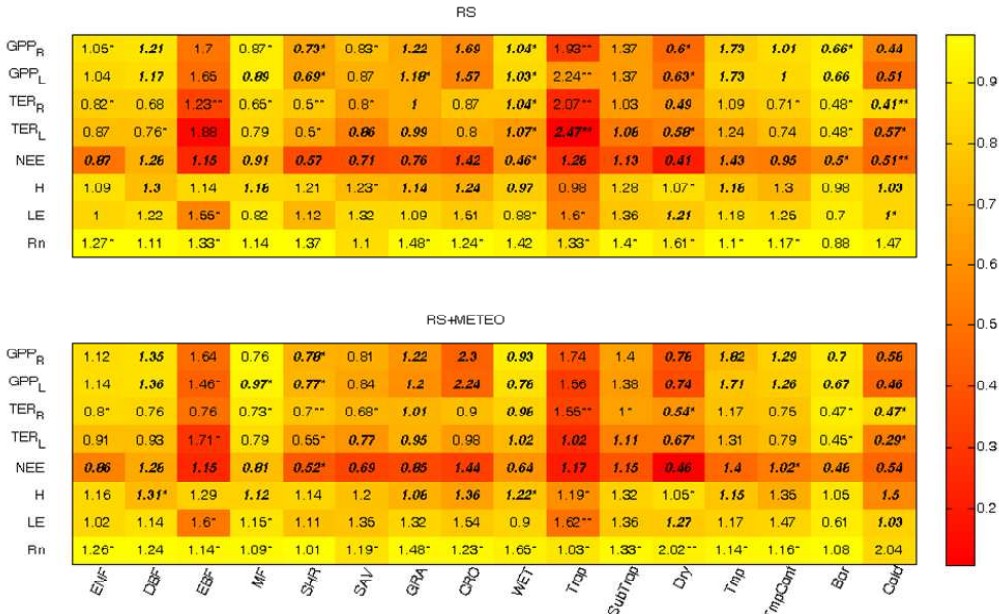

**Fig. 3**: Performance of FLUXCOM median estimates per climate zone and plant functional type (PFT). The colored matrices showed the median values of $R^2$ (red pixels for low $R^2$, yellow pixels for high $R^2$). Number indicated the RMSE (units of $CO_2$ fluxes were $gCm^{-2}d^{-1}$ and $MJm^{-2}d^{-1}$ in the case of energy fluxes). Oblique and bold font were used when the relative RMSE (normalized for the mean observed fluxes per PFT and climate zone) was greater than 0.5. The symbols '**' after RMSE were used when the weight of bias (estimated as the ratio between the square of median absolute bias and the MSE) was greater than 0.5, instead '*' symbols were used if the weight of bias was between 0.25 than 0.5. No symbols were used if the weight of bias is lesser than 0.25.

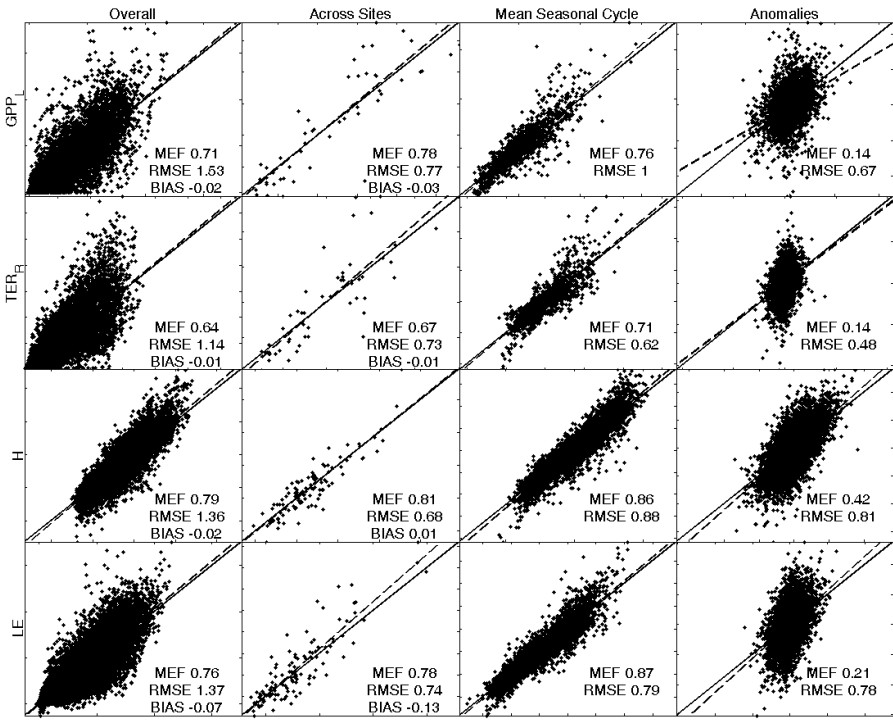





**Fig. B1**. Scatterplots of observed data by eddy covariance (y-axis) and the median ensemble of modeled fluxes by RS setup (x-axis). The
panels from left to right were the eight days predictions, the across sites variability, the mean seasonal cycle and the eight days anomalies.
The fluxes considered here were GPP$_L$ (first row), TER$_R$ (second row), H (third row) and LE (fourth row).

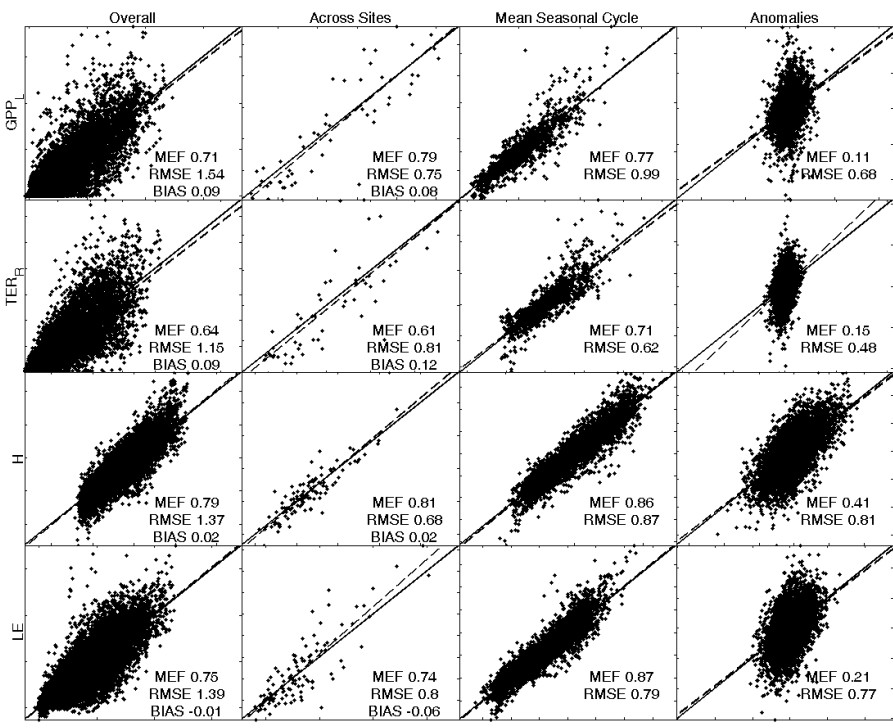

**Fig. B2**. As Fig. B1 but the predictions (x-axis) were obtained by the RS+METEO setup.