# Peer review of "S1 Eddy covariance study sites used for FLUXCOM experiment"

_Biogeosciences, 2015_

## Referee Comment (RC1) · Anonymous Referee #2 · 28 Mar 2016

In ''Predicting carbon dioxide and energy fluxes across global FLUXNET sites with regression algorithms" the authors cross-validate an ensemble of machine learning methods to document the performance of these methods in terms of their spatio-temporal performance. This study is very useful given the role of eddy covariance observations in land-atmosphere studies and the increasing importance of some of the upscaled EC-products in model validation and data analysis. In my opinion the study falls well within the scope of Biogeosciences and addresses a topic that is of interest to the journal's readership.

The work underlying the study is of high quality, however, the current presentation can be much improved. If the authors would try to separate the results and discussion,

it would become apparent that there is hardly any discussion. Despite the carefully worded objectives, the reader is left with a ''so-what'' feeling. The way the objectives are worded is too technical and is unlikely to excite many readers. That would be a pity as the results desirve better. Are you looking for the best method or do you want to quantify the upscaling uncertainty? Both perspectives could be of interest but from the conclusions I understood that all ML results will be archived and that the ensemble will be distributed. If my understanding is correct, this information should already be presented in the introduction. If this is indeed the context of the study, searching for the best ML method becomes less relevant but estimating spatio-temporal patterns in uncertainty becomes even more relevant as users may want to know the uncertainty of the ensemble mean. Also, the reader may want to know how much the uncertainty can be reduced by adding remote sensing and meteorological information in the up-scaling process. Listing the current limitations (saturation point) would be very useful, for example, is there anything to gain by adding meteorological data when upscaling NEE?

Several interesting findings are not further explored, for example, line 329 reads ''...suggested that the choice of the explanatory variable had higher impact than the choice of the ML technique for the pattern of predictions''. This is a very useful and important finding but it is not at all discussed. There are too many loose ends such as the paragraph on line 317 that reads '' Nonetheless, the differences between the experimental setups were less appreciable.'' A paragraph should have an introduction, a body and a concluding phrase signifying the implication of the result/discussion. This is often missing leaving it to the reader to guess what the authors want to say.

Both the structure and language of the manuscript could be improved. The authors choose to use their objectives to structure the paper. I find the objectives very technical and they seems to overlook some of the more interesting questions and answers the study could provide. As an alternative the manuscript could discuss the possibilities and limitations of spatial upscaling and then the possibilities and limitations of temporal

upscaling. Defining more general overarching objectives is likely going to result in a better structure and discussion. For the typos and grammar ask help from one of the three native speakers on the manuscript. It makes me wonder whether all co-authors even made the effort to read the manuscript.

The display items show a lot of information but not in a way that is easy to interpret or a way that at first sight supports the conclusions. The challenge of synthesis study such as this one is to summarize the information in easy to grasp figures and tables. In my opinion the authors failed in doing so. This issue is apparent from the first paragraph of the results where Table 3 is cited in support of the statement that '' The ensemble median estimate always outperformed the median performance of ML-specific methods" but the way I read this table it does not contain information of the specific methods. The detailed information could be moved to the appendices. Prepare figures that support the main message(s) of this study, for example, a figure that shows how some temporal characteristics are lost for certain fluxes and/or a figure/map that shows the regions where the methods diverge most.

––––––––––––––––––––––––––––

---

## Referee Comment (RC2) · Anonymous Referee #3 · 1 Apr 2016

Tramontana et al. present a study in which they have fit various empirical models to CO2 water and energy fluxes across eddy-covariance sites. The results are clear and unsurprising: the statistical fitting methods all performed comparably, and the energy fluxes were more easily predicted by the statistical models. The study is well executed and no doubt will be well cited by follow-on studies that use this dataset for research. That said, I was somewhat disappointed at the level of insight the results conveyed. It is not clear what we have learned beyond a statistical comparison of fits. The results are presented as dense tables of statistics (even the figures are graphical representations of statistical tables) where fits are classified as better or worse than others, but with little or no discussion or interpretation of the underlying biogeosciences.

[Figure]

The manuscript would clearly benefit from a more descriptive comparison of modeled vs. data. For example, I would suggest presenting Figures B1 and B2 in the main text. Perhaps see Mahecha et al. for ideas on how to gain more insight from comparisons of models and observations. Mahecha, M. D. et al. Comparing observations and process-based simulations of biosphere-atmosphere exchanges on multiple timescales. J. Geophys. Res. 115, G02003 (2010).

One important note is that GPP and RE are modeled. From the methods it appeared that gap-filled data were also included in the fitted data. Some discussion on comparing models with modeled data is merited.

The authors briefly reference observational uncertainty when considering their results but it is not clear to what extent they have accounted for uncertainty. Do the models fall within the uncertainty of the observations?

The main benefit of such regression algorithms in the context of Fluxnet is scaling. It would greatly increase the impact of the paper if the authors used the trained algorithms to scale each of the fluxes to the globe. This would be relatively easy to do, and the difference between the global estimates would be much more insightful than the statistics currently presented.

The manuscript would benefit from revisions for the correct use of English.

Minor comments:

Line 31: "ML and setups"?

Line 41: Updated 2013 IPCC reference

Line 44: "are equal"

Line 45: "accounted for"

Line 59: Perhaps cite Moffat et al. here, as it contains a good discussion of the relative benefits of both approaches. Moffat, A. M., Beckstein, C., Churkina, G., Mund, M.

& Heimann, M. Characterization of ecosystem responses to climatic controls using artificial neural networks. Glob. Chang. Biol. 16, 2737–2749 (2010).

Line 61: "generally come from"

Line 78: "The ML tools used span"

Line 105: So gap-filled data of high confidence are being included? Some discussion on the dangers of fitting a model to modeled data might be warranted.

Line 113: "we removed 5%"

Line 294: "with respect to"

Line 294-296: On what spatial and temporal scale? Daily NEE is typically not affected by external factors. The sentence reads as a result of the study but in reality it is a hypothesis you propose to explain the lack of model fit. You do not identify management influences or lagged effects.

Line 298: How were the uncertainties in H, LE and NEE quantified? I do not see that presented anywhere. It is not clear where your claim that the uncertainties were larger comes from.

Line 340: This sentence is not clear.

---

## Referee Comment (RC3) · Anonymous Referee #4 · 11 Apr 2016

General comments: 1) Good paper, but English can significantlly be improved. I added the reviewed mansucript with a lot of examples for improvement. Please take care of this action. 2) Use the present tence wherever possible. 3) Scientifically I have no comments on this paper. Its thesis is sound and the argumentation as well.

Specific comments: 1) Page 5 line 83-84: According to me, VI's are only partially descriptive for vegetation state! Please comment and discuss on my statement. 2) Page 8, line 138: Give references for the QA/QC standard procedure for flux post-processing. 3) Page 8, line 178: FPAR is an erroneous acronym for fAPAR. Please correct in the manuscript. 4) Page 9, line 189: Why was the Maximum Value Composite criterion (MVC) not used? Please explain. 5) Page 9, line 192-193: A

days composite? What criterion was used for this multitemporal composite? 6) Page 9, line 199: VPD? Define VPD please. How was it calculated? 7) Page 9, line 201-202: ERA-Interim dataset? Give references for this dataset. 8) Page 17, line 387: predictive skill. This is a rather nonsensical expression, rather use predictive capacity or capability. 9) Page 18, Line 411: Individual ML methods also exhibited higher skill than... What does skill mean here? Unclear to me.

Please also note the supplement to this comment:
http://www.biogeosciences-discuss.net/bg-2015-661/bg-2015-661-RC3-supplement.pdf

**Supplement:**

**Predicting carbon and energy fluxes across global FLUXNET sites with regression**

**algorithms**

G. Tramontana[1], M. Jung[2], G. Camps-Valls[3], K. Ichii[4,5], B. Raduly[1,6], M. Reichstein[2], C. R.

Schwalm[7], M. A. Arain[8], A. Cescatti[9], G. Kiely[10], L. Merbold[11], P. Serrano-Ortiz[12], S.

Sickert[13], S. Wolf[14] and D. Papale[1].

[revised manuscript text omitted]
 has advantages and disadvantages. While RS  provides products with  a spatial resolution  of 1 km or less, data are limited to the MODIS era (2000-present) and has a coarse (weekly) temporal resolution. The uncertainties of remote sensing data at tower locations due to finer scale spatial heterogeneity may also degrade the performance of the ML methods. RS+METEO can take advantage of information from meteorological variables, and is resistant to the noise of remote sensing time series because only mean seasonal cycle of data from satellite  are used. RS+METEO allows for upscaled products over a longer time period (because not constrained by the availability of MODIS data) and finer time scale (daily). However the predictive  capacity of this setting  is conditioned by missing  information regarding the interannual variability of vegetation greenness (phenology). In addition the use of meteorological gridded datasets introduces another source of uncertainty  e.g., potential dataset specific biases and  their typical coarse spatial resolution (0.5 degrees or larger).

**2.3.2. Variable selection**

Combining remote sensing and meteorological data (see Sect. 2.1.2 and 2.1.3) we created additional variables  model input. In the case of RS+METEO setup we derived the

Water Availability Index (WAI)  is based on a simple soil water balance model (for more details see supplementary material, Sect. S3) as an attempt to better represent waterstressed conditions. For both setups we derived proxies for absorbed radiation as the product between vegetation greenness (e.g. EVI, NDVI,  fAPAR) and available energy (e.g. daytime LST, Rg, and potential radiation). Other derived variables include the mean seasonal cycle (MSC) of dynamic variables (phenology) and associated metrics (minimum, maximum, amplitude, and mean). For remote sensing predictors, the MSC and associated metrics are based on the period 2001-2012 while  climate variables are based on the bias corrected daily long-term ERA-Interim data reference period (1989-2010). In total, 216

potential explanatory variables were created for RS and 231 for RS+METEO (see supplementary material S4 for details).

For each setup we selected a small subset of variables  optimally suitable to predict  target fluxes using a variable selection search algorithm. Variable selection is an important component in  spatial upscaling  since it improves the accuracy of predictions  while computational costs of  global predictions are minimized. We used the Guided Hybrid

Genetic Algorithm (GHGA) published by Jung and Zscheischler (2013), which was designed for variable selection problems with many candidate predictor variables and computationally expensive cost functions. The GHGA requires the training of a regression algorithm (here RF) to estimate the cost associated with selected variable subsets (see S5

for details).

Instead of doing a computationally demanding variable selection for each individual flux, variable selection runs were performed for the RS and RS+METEO setups and separately for carbon and energy fluxes. This procedure has the advantage that the resulting global products originate from a consistent set of predictor variables. The selected variables for the prediction of carbon and energy fluxes are listed in Table 2.

**2.3.3. Model training**

The capability of ML methods to spatially extrapolate carbon and energy fluxes has been evaluated by a 10-fold cross-validation strategy. The training datasets were stratified into

10-folds, each one roughly containing 10% of the data. Entire sites a  assigned to each fold (Jung et al., 2011). The target values for each fold have been  predicted based on the training using the remaining nine folds. Due to computational expense of the RS+METEO

setup, only one method representing each "family" – multiple regressions, RF, MARS,

ANN and KRR – was trained.

ML methods base settings have been  tuned before the training (for further details, see supplementary material S6). These hyper-parameters account for regularization to avoid overfitting, as well as for the shape and smoothness constraints. Instead, the model parameters were estimated for each ML every time in each fold .

**2.3.4. Model evaluation**

 To highlight the differences between the RS and RS+METEO setups, the daily output from RS+METEO has been  aggregated to eight days time step; the same periods and sites have been  used for  comparison. Besides the statistical analysis of the individual ML

cross-validation results, we focussed  on the ensemble median estimate,  defined as the median predicted value across all ML for a given setup and time step. The advantages of the median ensemble estimate is the robustness of the predictions for the contribution of many ML that reduce the risk of outliers  in the extrapolation exercise.

We used different metrics to evaluate the ML performance such as the Nash and Sutcliffe model efficiency (MEF), the root mean square error (MSE), the empirical BIAS, the

Pearson's linear correlation coefficient (ρ), the coefficient of determination ($R^2$) and the ratio of variance (ROV).

MEF (Nash and Sutcliffe, 1970), if the capability of a model to estimate a target variables is better than a reference model. If the reference model is the mean value of the target,

MEF can be calculated as:

$$MEF = 1 - \frac{\sum_{i=1}^{n}(x_i - y_i)^2}{\sum_{i=1}^{n}(y_i - \bar{y})^2}$$    (1)

where $x_i$ and $y_i$  are the predicted and the observed values respectively and $\bar{y}$ is the mean value of the observations (here the reference model). MEF can vary between -inf to

1; if MEF > 0 the predictive  capacity of the model is better than the mean (MEF = 1 for the ideal model), if MEF=0 the predictive  capacity of the model is equivalent to the mean, finally if

MEF < 0, the predictive  capacity of the mean value of the target is better than the model.

The RMSE  is estimated as the root square of the  values mean of the squared residuals:

$$RMSE = \sqrt{\frac{\sum_{i=1}^{n}(x_i - y_i)^2}{n}}$$    (2)

The BIAS  is evaluated as the differences between the mean values of model's residuals

$$BIAS = \frac{\sum_{i=1}^{n}(x_i - y_{i)}}{n}$$    (3)

Following Gupta et al. (2009) the importance of bias on the overall uncertainty  is evaluated as the ratio between the square of BIAS and Mean Square Error, the latter estimated as the square value of RMSE.

The Pearson's linear correlation coefficient (ρ)  is the ratio between the covariance between the modeled and observed values ($\sigma_{xy}$) and the product of the standard deviation of modeled ($\sigma_x$) and observed ($\sigma_y$) values:

$$\rho = \frac{\sigma_{xy}}{\sigma_x \sigma_y}$$  (4)

$R^2$  is estimated as the squared value of ρ; finally ROV  is evaluated as the ratio between predicted and observed standard deviation.

We evaluated the overall predictive  capacity of the models, evaluating the consistency among trained ML approaches and across the experimental setup. Then we evaluated the capability of the regression models to predict site-specific mean fluxes, mean seasonal cycle (MSC), and anomalies (Jung et al., 2011). The MSC per site was calculated using the averaged values for each eight days period across all available years, but only when at least two values (i.e., years) for each eight days period  are available. To assess the mean values of the study sites, we calculated the mean of the MSC if at least 50% of the

46 eight daily values  are present, whereas the weekly anomalies  have been calculated as the deviation of a flux value from the MSC. Finally, the mean site value was removed from the MSC to disentangle the seasonal variation from the mean, thus making the MSC and mean complementary.

We also analyzed the performance for the different Köppen climate zones and IGBP plant functional types (PFT's). In particular. we computed for each flux, setup and tower site the performances of ML median estimate. Then, for each setup, we estimated the median value of the site-by-site statistics per PFT and climate zone.

**3. Results and Discussions**

**3.1 Overview**

The ensemble median estimate always outperformed the median performance of ML- specific methods (the median value of metrics calculated for individual ML) (Table 3;

Appendix A). Individual ML methods also exhibited higher skill than multiple linear *(what does skill mean here? Unclear phrasing)*

regressions (higher MEF and lower RMSE; Fig. 1). This highlights the added value of ML

methods as these are able to account for nonlinearities in either explanatory variables or fluxes. Overall, using the ensemble median estimate gives a representative overview of

ML-based flux predictions.

**3.2 Predictive  capacity of carbon and energy fluxes**

Predictive  capacity of the ensemble median estimate clustered into tiers whereby energy fluxes are uniformly better predicted than carbon fluxes: Rn > H/LE/GPP > TER > NEE

(Table 3). The highest  predicitve capacity levels as exhibited by net radiation shows near perfect agreement; Rn displays a model efficiency (MEF) of 0.91-0.92 and a correlation of 0.96.

The decline in  predicitve capacity for the second tier fluxes is ca. 15% to 20%; MEF for H, LE, and GPP is

0.79, 0.75-0.76, and 0.71 respectively. The lowest two tiers exhibits 20% and 40%

declines in MEF (0.57-0.64 and 0.43-0.46 for TER and NEE respectively). These relative rankings are unchanged regardless of skill metric used—apart from RMSE where the difference in fluxes units and magnitude, confounds a direct comparison (Table 3)—

suggesting that accuracy and precision scale linearly.

There were only minor performance differences between the two carbon fluxes partitioning methods (Table 3), although for the RS setup, the performance of $TER_L$ were comparatively lower than $TER_R$ (lower MEF, ρ and ROV). A similar trend was not found in the case of RS+METEO setup.

The overall  *predicitve capacity* profile in this study confirms previous upscaling efforts (Jung et al., 2011;

Yuan et al., 2010). This relatively stable cross-study  *predicitve capacity* gradient reflects the information content of the available predictor variables. The spatiotemporal variability of remotely sensed land surface properties is well-suited to predict the top tier fluxes (Rn, H, LE, and

GPP) (Jung et al., 2008; Tramontana et al., 2015; Xiao et al., 2010;.Yang et al., 2007)

The higher  *predicitve capacity* associated with energy fluxes suggests that these fluxes are more easily predictable using the drivers selected in particular respect to NEE. In fact NEE is strongly controlled by external factors such *as* management and disturbances (Amiro et al., 2010;

Thornton et al., 2002) and by lag and memory effects (Bell et al., 2012; Frank et al., 2015,

Papale et al., 2015; Paruelo et al., 2005), which are both poorly captured by predictor variables typically used in upscaling and poorly constrained in general, i.e., data limited.

Another reason for the low performances in NEE simulation can be in the uncertainties in the measurements that are larger compared to H and LE and have an important effect being NEE the difference between two large components (GPP and TER).

Among the carbon fluxes, GPP is  *the best* predicted probably because the seasonal cycle and canopy properties, which  *are* strongly related to GPP,  *are* well represented by the

ML drivers. The intermediate skill of TER, relative to carbon fluxes only, is supported by its tight coupling to the well-predicted GPP and the availability of predictor variables that capture the temperature dependency of respiration. However specific drivers for TER could be still missing. In fact in contrast to GPP, the canopy properties are less important drivers of TER, while the soil properties, carbon pools and their turnover rates are key for respiratory processes (Amiro et al., 2010) but not available to be used as drivers. This likely explains the poor performance for TER in comparison with GPP.

**3.3 Are  flux predictions consistent between RS and RS+METEO?**

 *predicitve capacity*, in terms of both performance tiers and absolute value of skill metrics, are similar for both RS and RS+METEO approaches with some differences, in particular: (1) RS and RS+METEO diverge more for those fluxes and sho lower overall  *predicitve capacity* levels, in particular for NEE (Fig. 1, Table 3); (2) MEF and correlation values  *are* slightly larger for RS than RS+METEO, excluding $TER_L$ where the opposite was found, indicating an important role of the meteorological data for this version of the ecosystem respiration. It should be considered that the differences in performance *can*  *be*  also due to a different ensemble size, with the RS composed of 11 individual ML-based ensemble members, whereas RS+METEO is based on only four. The overall good performance of the RS setup implies that carbon and energy fluxes can be mapped exclusively based on remotely sensed inputs allowing high-spatial resolution products and reduction of uncertainty due to the meteorological drivers spatialization (Tramontana et al., 2015). Nonetheless, the differences between the experimental setups were less appreciable.

**3.4 How different are the predictions of the various ML algorithms?**

Pair-wise $R^2$ values for model outputs (Table 4)  are close to unity ($R^2 \geq 0.90$), regardless of  experimental setup, with NEE showing a slightly lower value ($R^2 = 0.84$).  Between corresponding model residuals (Table 4), $R^2$ values ranged from 0.79 (Rn) to 0.89 (TER$_L$). Comparing the same ML technique but using different experimental setups (Table 4, RS vs. RS+METEO) show a similarly high, albeit somewhat diminished level of consistency ($R^2$ range  from 0.71 to 0.80 for model residuals). These finding highlight that the ML methods  a mapping between explanatory variables and target fluxes both reliably and robustly.  Between all three consistency checks there  is also a tendency for better predicted fluxes (e.g., H) to exhibit higher pair-wise $R^2$ values than poorly predicted fluxes (e.g., NEE). This is expected as more robust patterns—and therefore those that lead to greater predictive  power—are easier to extract regardless of ML algorithm and experimental setup  thus increasing consistency.  This broad consistency confirms that the extracted patterns are robust. The decline in $R^2$ when comparing the same ML trained with different drivers (RS vs RS+METEO) with respect to the correlation among ML methods with the same drivers, suggests that the choice of the explanatory variable has a higher impact than the choice of the ML technique for the pattern of predictions.

**3.5 How does  performance differ  between capturing  across-sites, seasonal and  deviations from  mean seasonal cycle variability?**

Decomposing FLUXNET data into across-sites variability, mean seasonal cycle, and interannual variability components (Sect. 2.3.4) reveals clear gradients in predictive capacity (Table 5 and Fig. 2). Across-sites variability is in general well-captured by the ML ($R^2$ range: 0.61 to 0.81 except for NEE) and the best predicted pattern for GPP and TER. This suggests that ML is suitable to reproduce the spatial pattern of mean annual fluxes. The variability in the mean seasonal cycle (at weekly time scale) was also uniformly well predicted ($R^2$ between 0.67-0.77 for GPP and TER, and between 0.86-0.98 for the energy fluxes) and the best predicted pattern for energy fluxes in particular for LE and Rn. The importance of phenology in carbon and energy annual fluxes is known (Joiner et al., 2014; Jung et al., 2011; Merbold et al., 2009; Wolf et al., 2011). Biases in its phenological dynamics (e.g. in the growing season length) can lead to biases in the predicted fluxes (Ichii et al., 2010). A clear benefit of the ML upscaling approach is that none of the parameters controlling phenology are prescribed, reducing the possibility of bias.

In contrast, interannual variability is generally poorly captured by all the ML approaches Only H and Rn showing an $R^2$ greater than 0.4. This low predictive capacity holds regardless whether weekly, monthly (Jung et al., 2011), or annual time steps are used (data not shown). This is likely due to a combination of missing predictor variables (e.g. disturbances, management, legacy effects) and the noise/uncertainty in both predictor and target variables playing a major role when small differences (like in the interannual variability) are predicted. The slightly better performances when sensible heat flux is estimated could be due to the lower uncertainty for this flux with respect to the others (only one sensor used, the sonic anemometer, in contract with the other fluxes where also the gas analyzer is used) but also to the fact that it is strongly and directly related to LST as a driver. In any case, predicting interannual variability remains one of the largest challenges in the context of empirical upscaling.

NEE is confirmed to be the most difficult and consequently poorest predicted flux (Table 3). Predicitive capacity for NEE is considerably lower relative to the other fluxes for between-sites variability ($R^2 = 0.46$), the mean seasonal cycle ($R^2 = 0.59$), and interannual variability ($R^2 = 0.13$, $TER_L$ is lowest at 0.10).

**3.6 How does the performance differ among climate zones or ecosystem types?**

Using climate zone and plant functional type (PFT) to disaggregate ML methods performances we find that in general energy fluxes are better predicted than carbon fluxes between the different climate zones and PFTs (Fig. 3 and Appendix C). The median $R^2$ between simulation and observation for carbon fluxes (excluding NEE) is larger than 0.6 for more than 75% of the PFT's and climate zones, while for the energy fluxes an $R^2$ greater than 0.7 is found for more than the 85% of the PFT's and climate zones (in all sites for Rn). NEE is 
[revised manuscript text omitted]

than 0.5. No symbols are used if the weight of bias is lesser than 0.25.

[Figure]

**Fig. B1**. Scatterplots of observed data by eddy covariance (y-axis) and the median ensemble of modeled fluxes by RS setup (x-axis). The panels from left to right are the eight days predictions, across sites variability, mean seasonal cycle and weekly anomalies. The fluxes considered here are

GPP$_L$ (first row), TER$_R$ (second row), H (third row) and LE (fourth row).

[Figure]

**Fig. B2**. As Fig. B1 but the predictions (x-axis) were obtained by the RS+METEO setup.

---

## Author Comment (AC1) · 31 May 2016

Dear Referee, Thanks very much for providing detailed comments to our work. Please find enclosed the responses to all comments point-by-point.

Comment 1: In "Predicting carbon dioxide and energy fluxes across global FLUXNET sites with regression algorithms" the authors cross-validate an ensemble of machine learning methods to document the performance of these methods in terms of their spatio-temporal performance. This study is very useful given the role of eddy covariance observations in land-atmosphere studies and the increasing importance of some of the upscaled EC-products in model validation and data analysis. In my opinion the study falls well within the scope of Biogeosciences and addresses a topic that is of

interest to the journal's readership. The work underlying the study is of high quality, however, the current presentation can be much improved. If the authors would try to separate the results and discussion, it would become apparent that there is hardly any discussion. Despite the carefully worded objectives, the reader is left with a "so-what" feeling. The way the objectives are worded is too technical and is unlikely to excite many readers. That would be a pity as the results deserve better.

Reply 1: We thank the reviewer for his positive comments on the quality and relevance of our work and more important for the critical view of the structure. We agree with the reviewer that an improved presentation of the material will make the paper more exciting and we will follow the reviewer's suggestion to frame the objectives of the paper in an attractive way for a broader audience. We will also insert a paragraph in the introduction to clarify that this paper presents the backbone of an ensemble of global gridded flux products generated by the FLUXCOM initiative (which will be introduced in follow-up manuscripts). We only partially agree with the reviewer that there is hardly any discussion – there is substantial discussion in particular related to methodological aspects, which play an essential role in the paper. However we take the point and we are going to add some discussion points concerning biogeosciences topics and limitation of proposed approach that, we agree, are currently missing. We will carefully consider to separate Results and Discussion in the revised version of the manuscript

Comment 2: Are you looking for the best method or do you want to quantify the up-scaling uncertainty? Both perspectives could be of interest but from the conclusions I understood that all ML results will be archived and that the ensemble will be distributed. If my understanding is correct, this information should already be presented in the introduction. If this is indeed the context of the study, searching for the best ML method becomes less relevant but estimating spatio-temporal patterns in uncertainty becomes even more relevant as users may want to know the uncertainty of the ensemble mean.

Reply 2: We inserted a paragraph in the Introduction to clarify that the cross-validation experiment presented in this manuscript is part of a project that aims to deploy an ensemble of globally upscaled fluxes (CO2 and energy) using data-driven models (FLUX-COM). As correctly pointed out by the reviewer, most users will be interested in using the ensemble median of different machine learning (ML) methods; therefore, we focused the paper on the performance of the median ensemble. This point will be further clarified in the new Introduction section. A section showing also the consistency among predictions by different machine learning methods is currently presented (section 3.4) but its relevance to the aims of the manuscript is marginal. We will likely move it to the supplementary material to make the manuscript more concise.

Comment 3: Also, the reader may want to know how much the uncertainty can be reduced by adding remote sensing and meteorological information in the upscaling process.

Reply 3: This is an interesting and important question for which we have designed the two different experimental set-ups. It is a particularly important question because the use of information from in-situ measured meteorological data implies a trade-off with introducing additional uncertainty inherent the gridded meteorological data needed for the global flux products (as discussed in different places in the manuscript). The comparison between ML based on only satellite drivers and in situ meteorological ones is a key point of our manuscript. We will sharpen this aspect in the appropriate places of the manuscript (Introduction and Discussion) which we believe will satisfy the reviewer's comment.

Comment 4: Listing the current limitations (saturation point) would be very useful, for example, is there anything to gain by adding meteorological data when upscaling NEE?

Reply 4: Identifying and discussing the limitations of machine learning based upscaling is the overarching objective of this manuscript and the Results and Discussion sections are dedicated to that with the main findings summarized in the final section of the paper. Reviewer 2 raises an interesting point about possible saturation points in the predictions and we are working to add a section or a paragraph on this specific aspect. Detecting possible saturation points in the predictions is however not trivial. For example, the uncertainties of the measured fluxes grow with their magnitude such that there will always be observed points that are above the largest predicted value.Other limitations are due to the nature of some of the machine learning used in the ensemble (e.g. Random Forest or Artificial Neural Network) that generally do not provide output values outside their training domains. All this will be discussed in the new manuscript version.

Comment 5: Several interesting findings are not further explored, for example, line 329 reads ''suggested that the choice of the explanatory variable had higher impact than the choice of the ML technique for the pattern of predictions''. This is a very useful and important finding but it is not at all discussed. There are too many loose ends such as the paragraph on line 317 that reads '' Nonetheless, the differences between the experimental setups were less appreciable.'' A paragraph should have an introduction, a body and a concluding phrase signifying the implication of the result/discussion. This is often missing leaving it to the reader to guess what the authors want to say. Both the structure and language of the manuscript could be improved.

Reply 5: We thank the reviewer for identifying this problem; in the revised manuscript we will address these issues with the help of a proofreader native speaker to help readability and avoid loose send sentences.

Comment 6: The authors choose to use their objectives to structure the paper. I find the objectives very technical and they seems to overlook some of the more interesting questions and answers the study could provide. As an alternative the manuscript could discuss the possibilities and limitations of spatial upscaling and then the possibilities and limitations of temporal upscaling. Defining more general overarching objectives is likely going to result in a better structure and discussion.

Reply 6: We agree that the original version of the manuscript had a very technical structure. In the revised manuscript we will place the objectives of the paper in the

frame of more general questions that are relevant for a broader audience, in particular to potential users of the global products generated by FLUXCOM. We need however to keep the technical depth and precision for 'good scientific practice' since the methodology presented here is the basis for global flux products. To improve readability we will move some more technical details to the supplementari information (SI). We thank the reviewer for the suggestion on how to split the discussion. We carefully thought about splitting it into 'spatial' and 'temporal' upscaling. We came to the conclusion that structuring the discussion in 'methodological' and 'biogeochemical' questions is more appropriate and appealing.

Comment 7: For the typos and grammar ask help from one of the three native speakers on the manuscript. It makes me wonder whether all co-authors even made the effort to read the manuscript. The display items show a lot of information but not in a way that is easy to interpret or a way that at first sight supports the conclusions. The challenge of synthesis study such as this one is to summarize the information in easy to grasp figures and tables. In my opinion the authors failed in doing so. This issue is apparent from the first paragraph of the results where Table 3 is cited in support of the statement that '' The ensemble median estimate always outperformed the median performance of ML-specific methods" but the way I read this table it does not contain information of the specific methods.

Reply 7: The revised manuscript will be proofread and edited by a native speaker with a focus on distilling the volume of information into a coherent storyline that supports the conclusions. In addition the final papers are proofread also by the journal. We will ensure a high quality of figures and tables that convey the main messages and reference them at the appropriate places in the text.

Comment 8: The detailed information could be moved to the appendices. Prepare figures that support the main message(s) of this study, for example, a figure that shows how some temporal characteristics are lost for certain fluxes and/or a figure/map that shows the regions where the methods diverge most.
Reply 8: Thank you for suggestion. As outlined before, the overall presentation of the material will be substantially modified in the revised version. We prefer not add maps showing the uncertainty at global scale because this is subject of another manuscript (in preparation) on the global products but we will add figures bringing the same type of message.

———————————————————

---

## Author Comment (AC2) · 31 May 2016

Dear Referee, Thanks very much for providing detailed comments to our work. Please find enclosed the responses to all comments point-by-point.

Comment 1: Tramontana et al. present a study in which they have fit various empirical models to CO2 water and energy fluxes across eddy-covariance sites. The results are clear and unsurprising: the statistical fitting methods all performed comparably, and the energy fluxes were more easily predicted by the statistical models. The study is well executed and no doubt will be well cited by follow-on studies that use this dataset for research. That said, I was somewhat disappointed at the level of insight the results conveyed. It is not clear what we have learned beyond a statistical comparison of fits. The

results are presented as dense tables of statistics (even the figures are graphical representations of statistical tables) where fits are classified as better or worse than others, but with little or no discussion or interpretation of the underlying biogeosciences.

Reply 1: We thank the reviewer for these comments and we agree that results and discussions should be restructured in order to bring a more interesting and clear message. We will also revise the Introduction section to frame our work in more broad and relevant questions and dedicate space to discussion about how this work is relevant for answering ecological questions.

Comment 2: The manuscript would clearly benefit from a more descriptive comparison of modeled vs. data. For example, I would suggest presenting Figures B1 and B2 in the main text. Perhaps see Mahecha et al. for ideas on how to gain more insight from comparisons of models and observations. Mahecha, M. D. et al. Comparing observations and process-based simulations of biosphere-atmosphere exchanges on multiple timescales. J. Geophys. Res. 115, G02003 (2010).

Reply 2: We thank Referee #3 for the suggestion, but we think that the paper would lose focus when including a too detailed site-by-site analysis as was presented in Mahecha et al. We will restructure and improve the presentation of the results and discussion sections and we consider to incorporate figures B1 and B2 in the main text as suggested by the reviewer.

Comment 3: One important note is that GPP and RE are modeled. From the methods it appeared that gap-filled data were also included in the fitted data. Some discussion on comparing models with modeled data is merited.

Reply 3: It is correct that GPP and RE are not direct measurements but are derived using models where model parameters are estimated in short temporal moving windows. To acknowledge this source of uncertainty we employed GPP and RE estimates from two independent flux partitioning methods. The first extrapolates daytime ecosystem respiration using fitted relationships on the basis of nighttime data (where RE is

measured due to the absence of GPP) whereas the second uses daytime NEE and an hyperbolic light response curve to derive GPP and RE. Both methods yield highly consistent results. The difference between flux partitioning methods turns out to be even smaller than the spread across ML algorithms. We plan to insert a paragraph in the discussion section on this aspect and the model-to-model comparison issue. Regarding the fact that data are gap-filled, we filtered the data and periods with more than 20% of gap filled data with low confidence were not used in upscaling. As such the influence of gap filling was minimized. Restricting the training data set of the ML methods to periods with 100% of measured fluxes is impossible because almost no data would be available at the time resolution used. It is also important to consider that the gap filling algorithm utilizes highly localized and site specific relationships between fluxes and meteorological conditions (MDS method, Reichstein et al 2005), while the ML cross-validation presented in the paper are based only on data from other sites. We will insert a paragraph in the discussion on this 'data quantity vs data quality' trade-off.

Comment 4: The authors briefly reference observational uncertainty when considering their results but it is not clear to what extent they have accounted for uncertainty. Do the models fall within the uncertainty of the observations?

Reply 4: We did not account for the propagation of the measurement uncertainties in a formal way; however, we will add a paragraph in the discussion about this issue. The random uncertainty of fluxes can be thoroughly quantified but we guess it does not have a big impact because: (a) it diminishes quickly as one aggregates to daily or even eight days values, and (b) the risk of model's bias is reduced with random uncertainties. The bigger problem is related to the systematic uncertainties for which we only have some heuristic approaches to assess them (u*, different flux partitioning methods, energy balance closure . . .). We will add a sentence explaining that a rigorous quantification of measurement uncertainties, both random and systematic, would allow for propagating those formally with some of the machine learning methods used.

Comment 5: The main benefit of such regression algorithms in the context of Fluxnet is

scaling. It would greatly increase the impact of the paper if the authors used the trained algorithms to scale each of the fluxes to the globe. This would be relatively easy to do, and the difference between the global estimates would be much more insightful than the statistics currently presented.

Reply 5: We agree this would add significant value in the context of scaling FLUXNET. A companion paper that uses our results as a point of departure is under preparation. It will feature global estimates as well as wall-to-wall maps.

Comment 6: The manuscript would benefit from revisions for the correct use of English. Minor comments: Line 31: "ML and setups"?

Reply 6: Different ML and experimental setups. If ML comparison will be moved in supplementary materials, we can leave only experimental setups.

Comment 7: Line 41: Updated 2013 IPCC reference, Line 44: "are equal" Line 45: "accounted for" Line 59: Perhaps cite Moffat et al. here, as it contains a good discussion of the relative benefits of both approaches. Moffat, A. M., Beckstein, C., Churkina, G., Mund, M. &Heimann, M. Characterization of ecosystem responses to climatic controls using artificial neural networks.Glob.Chang.Biol.16, 2737–2749 (2010). Line 61: "generally come from" Line 78: "The ML tools used span" Line 113: "we removed 5%" Line 294: "with respect to"

Reply 7: Thank you, the point outlined in minor comments will be changed as suggested.

Comment 8: Line 105: So gap-filled data of high confidence are being included? Some discussion on the dangers of fitting a model to modeled data might be warranted.

Reply 8: We are going to add discussion on this issue in the methodological related discussion.

Comment 9: Line 294-296: On what spatial and temporal scale? Daily NEE is typically not affected by external factors. The sentence reads as a result of the study but in

reality it is a hypothesis you propose to explain the lack of model fit. You do not identify management influences or lagged effects.

Reply 9: Yes. We will clarify it in the discussion of the revised manuscript

Comment 10: Line 298: How were the uncertainties in H, LE and NEE quantified? I do not see that presented anywhere. It is not clear where your claim that the uncertainties were larger comes from.

Reply 10: We did not account for the measurement uncertainties in a formal way.

Comment 11: Line 340: This sentence is not clear

Reply 11: Sorry for the misunderstanding. The sentence will be removed from the new discussion section.

---

## Author Comment (AC3) · 31 May 2016

Dear Referee, Thanks very much for providing detailed comments to our work. Please find enclosed the responses to all comments point-by-point.

Comment 1: General comments: 1) Good paper, but English can significantly be improved. I added the reviewed manuscript with a lot of examples for improvement. Please take care of this action.

Reply 1: We thank the reviewer for his/her comments. We will improve the English in the revised manuscript by the help of a native speaker and following the suggestion by the reviewer as proposed in the supplementary material.

[Figure]

Comment 2: 2) Use the present tense wherever possible.

Reply 2: Thank you for suggestion. We are open to this suggestion: the use the past tense was requested by the managing editor but we are ready to change if needed.

Comment 3: 3) Scientifically I have no comments on this paper. Its thesis is sound and the argumentation as well. Specific comments: 1) Page 5 line 83-84: According to me, VI's are only partially descriptive for vegetation state! Please comment and discuss on my statement. Reply 3: Yes, we agree with the reviewer. We will clarify this point in the revised manuscript.

Comment 4: 2) Page 8, line 138: Give references for the QA/QC standard procedure for flux post-processing.

Reply 4: Thanks, we will add the appropriate references.

Comment 5: 3) Page 8, line 178: FPAR is an erroneous acronym for fAPAR. Please correct in the manuscript.

Reply 5: Thank you, we will correct it.

Comment 6: 4) Page 9, line 189: Why was the Maximum Value Composite criterion (MVC) not used? Please explain.

Reply 6: We have used different MODIS product, each one having specific composite method. The composite methods have been explained in the reference papers of the MODIS products (they are reported in the manuscript text). About the specific point outlined by the referee, firstly we have filtered the good quality data on the basis of the MODIS quality check layer, then we have extracted the mean value of a 3X3km2 area centered on the tower location to better represent the flux tower footprint, as reported in Xiao et al. (Xiao, J. ,et al: A continuous measure of gross primary production for the conterminous United States derived from MODIS and AmeriFlux data, Remote Sens Environ, 114, 576–591, doi: 10.1016/j.rse.2009.10.013, 2010).

Comment 7: 5) Page 9, line 192-193: A 16 days composite? What criterion was used for this multitemporal composite?

Reply 7: This is the standard MODIS composition system as described in Huete et al (2002). We will better clarify this in the reviewed version of the manuscript.

Comment 8: 6) Page 9, line 199: VPD? Define VPD please. How was it calculated?

Reply 8: We will define the acronym VPD (vapor pressure deficit) in the revised manuscript.

Comment 9: 7) Page 9, line 201-202: ERA-Interim dataset? Give references for this dataset.

Reply 9: We currently have used as reference "Dee, D. P., et al.: The ERA-Interim reanalysis: configuration and performance of the data assimilation system, Q.J.R. Meteorol Soc, 137, 553–597, doi: 10.1002/qj.828, 2011". To clarify this we can move (Dee et al., 2011) right after 'ERA-Interim'

Comment 10: 8) Page 17, line 387: predictive skill. This is a rather nonsensical expression, rather use predictive capacity or capability.

Reply 10: Thank you, we will try to use the suggested expression among the ones proposed by the reviewer.

Comment 11: 9) Page 18, Line 411: Individual ML methods also exhibited higher skill than... What does skill mean here? Unclear to me. Please also note the supplement to this comment:

Reply 11: Thank you. We are referring to the predictive capacity. We will use another expression from the ones proposed by the reviewer.

---

## Author Response (AR1)

*Dear Editor,*

*Thanks very much for providing the revision process to our work. Please find enclosed the revised version of the paper, and the responses to all reviewers comments. We have given full response to all Reviewers comments, and highlighted the responses in blue in the new version of the manuscript in order to expedite the revision process. In addition to the modifications suggested by the reviewers, we also improved some parts of the manuscript, rephrased some sentences for the sake of clarity and the quality of the figures. We think that the raised comments by the reviewers have been addressed and now the paper meets the quality standards of the Journal.*

*Kind regards,*

*The Authors*

**Anonymous Referee #2**

*Dear Referee,*
*Thanks very much for providing detailed comments to our work. Please find enclosed the responses to all reviewer's comments. We have given full response to all Reviewer's comments, and highlighted the responses in **blue***

Comment 1: In ''*Predicting carbon dioxide and energy fluxes across global FLUXNET sites with regression algorithms*'' the authors cross-validate an ensemble of machine learning methods to document the performance of these methods in terms of their spatio-temporal performance. This study is very useful given the role of eddy covariance observations in land-atmosphere studies and the increasing importance of some of the upscaled EC-products in model validation and data analysis. In my opinion the study falls well within the scope of Biogeosciences and addresses a topic that is of interest to the journal's readership. The work underlying the study is of high quality, however, the current presentation can be much improved. If the authors would try to separate the results and discussion, it would become apparent that there is hardly any discussion. Despite the carefully worded objectives, the reader is left with a ''so-what'' feeling. The way the objectives are worded is too technical and is unlikely to excite many readers. That would be a pity as the results deserve better.

*Reply 1: We thank the reviewer for his positive comments on the quality and relevance of our work and more important for the critical view of the structure. We agree with the reviewer that an improving of the manuscript's presentation was needed to make it more exciting and we have followed the reviewer's suggestion to frame the objectives in an attractive way for a broader audience (ln 81-91). We have also inserted a paragraph in the conclusion to clarify that this paper presents the backbone of an ensemble of global gridded flux products (Ln 424-425) generated by the FLUXCOM initiative (which it has been introduced in the manuscript's Introduction). We only partially agree with the reviewer that there was hardly any discussion – there was substantial discussion in particular related to methodological aspects, which have played an essential role in the paper. However we have taken the point and we have added some discussion points concerning biogeosciences topics and limitation of proposed approach that, we agree, were missing. We have also separated Results and Discussion in the revised version of the manuscript.*

Comment 2: Are you looking for the best method or do you want to quantify the upscaling uncertainty? Both perspectives could be of interest but from the conclusions I understood that all ML results will be archived and that the ensemble will be distributed. If my understanding is correct, this information should already be presented in the introduction. If this is indeed the context of the study, searching for the best ML method becomes less relevant but estimating spatio-temporal patterns in uncertainty becomes even more relevant as users may want to know the uncertainty of the ensemble mean.

*Reply 2: We have inserted a paragraph in the Introduction to clarify that the cross-validation experiment presented in this manuscript is part of a project that aims to deploy an ensemble of globally upscaled fluxes ($CO_2$ and energy) using data-driven models (FLUXCOM). As correctly pointed out by the reviewer, most users could be interested in using the ensemble median of different machine learning methods; therefore, we focused the paper on the performance of the median ensemble. This point was clarified in the new Introduction section as follows (ln 84-85 in the manuscript):*

***"We focused in particular on the ensemble median prediction because the ensemble median global product will likely be used extensively".***

*As consequence, the section showing the consistency among predictions by different machine learning methods were largely reduced in the new results section as follows (ln 287-289):*

*"However, the output provided by MLs methods showed high overall consistency among them, that increased when predictions were obtained by different MLs trained with the same experimental setup (RS else RS+METEO; for more details see Appendix B and Table B1)"*
*and the detailed contents moved in Appendix B.*

Comment 3: Also, the reader may want to know how much the uncertainty can be reduced by adding remote sensing and meteorological information in the upscaling process.

*Reply 3: This is an interesting and important question for which we have designed the two different experimental set-ups. It is a particularly important question because the use of information from in-situ measured meteorological data implies a trade-off with introducing additional uncertainty inherent the gridded meteorological data needed for the global flux products (as discussed in different places in the manuscript). The comparison between ML based on only satellite drivers and in situ meteorological ones is a key point of our manuscript and it was targeted discussed in a (new) discussion section of the revised manuscript focused on the ML experimental setup comparison (Section 4.1 Comparison between experimental setups, ln 323-350).*

Comment 4: Listing the current limitations (saturation point) would be very useful, for example, is there anything to gain by adding meteorological data when upscaling NEE?

*Reply 4: Identifying and discussing the limitations of machine learning based upscaling is the overarching objective of this manuscript and the Discussion sections are dedicated to that. Reviewer 2 raises an interesting point about possible saturation points in the predictions. Detecting possible saturation points in the predictions is however not trivial. For example, the uncertainties of the measured fluxes grow with their magnitude such that there will always be observed points that are above the largest predicted value. Other limitations could be due to the training with noisy drivers (it reduces the sensitivity of the trained algorithm); the noise is higher in the variables obtained by satellite (hence in the RS setup) but we did not find significant difference between the two experimental setup (maybe because the drivers were objectively selected and drivers selection optimized). All limitations of empirical upscaling by ML have been largely discussed in the Discussion section of the revised manuscript.*

Comment 5: Several interesting findings are not further explored, for example, line 329 reads ''suggested that the choice of the explanatory variable had higher impact than the choice of the ML technique for the pattern of predictions". This is a very useful and important finding but it is not at all discussed.
There are too many loose ends such as the paragraph on line 317 that reads '' Nonetheless, the differences between the experimental setups were less appreciable." A paragraph should have an introduction, a body and a concluding phrase signifying the implication of the result/discussion. This is often missing leaving it to the reader to guess what the authors want to say. Both the structure and language of the manuscript could be improved.

*Reply 5: We thank the reviewer for identifying this problem; we addressed these issues with the help of a proofreader native speaker that improved readability and avoided loose ends sentences.*

Comment 6: The authors choose to use their objectives to structure the paper. I find the objectives very technical and they seems to overlook some of the more interesting questions and answers the study could provide. As an alternative the manuscript could discuss the possibilities and limitations of spatial upscaling and then the possibilities and limitations of temporal upscaling. Defining more general overarching objectives is likely going to result in a better structure and discussion.

*Reply 6: We agree that the original version of the manuscript had a very technical structure. In the revised manuscript we have placed the objectives of the paper in the frame of more general questions that are*

*relevant for a broader audience, in particular to potential users of the global products generated by FLUXCOM. We need however to keep the technical depth and precision for 'good scientific practice' since the methodology presented here is the basis for global flux products. We thank the reviewer for the suggestion on how to split the discussion. We carefully have thought about splitting it into 'spatial' and 'temporal' upscaling. We came to the conclusion that discussing the 'methodological' and 'biogeochemical' questions of fluxes uscaling was more appropriate and appealing.*

Comment 7: For the typos and grammar ask help from one of the three native speakers on the manuscript. It makes me wonder whether all co-authors even made the effort to read the manuscript. The display items show a lot of information but not in a way that is easy to interpret or a way that at first sight supports the conclusions. The challenge of synthesis study such as this one is to summarize the information in easy to grasp figures and tables. In my opinion the authors failed in doing so. This issue is apparent from the first paragraph of the results where Table 3 is cited in support of the statement that '' The ensemble median estimate always outperformed the median performance of ML-specific methods" but the way I read this table it does not contain information of the specific methods.

*Reply 7: The revised manuscript has been proofread and edited by a native speaker with a focus on distilling the volume of information into a coherent storyline that supported the conclusions. In addition the final papers will be proofread also by the journal. We have added two figures (fig. 3a and 3b) and moved a table in appendix (currently the results section contain only two tables). Now the results and discussion section are easy to read and figures and tables support the main message of the manuscript. We do not want remove the 'detailed' table in the appendix, because they could be useful for the users of global products.*

Comment 8: The detailed information could be moved to the appendices. Prepare figures that support the main message(s) of this study, for example, a figure that shows how some temporal characteristics are lost for certain fluxes and/or a figure/map that shows the regions where the methods diverge most.

*Reply 8: Thank you for suggestion. As outlined before, the overall presentation of the material was substantially modified in the revised version. We have preferred not add maps showing the uncertainty at global scale because this is subject of another manuscript (in preparation) on the global products but the current figures brings the same type of message (e.g. Fig. 4).*

**Anonymous Referee #3**

*Dear Referee,*
*Thanks very much for providing detailed comments to our work. Please find enclosed the responses to all comments point-by-point.*

Comment 1: Tramontana et al. present a study in which they have fit various empirical models to $CO_2$ water and energy fluxes across eddy-covariance sites. The results are clear and unsurprising: the statistical fitting methods all performed comparably, and the energy fluxes were more easily predicted by the statistical models. The study is well executed and no doubt will be well cited by follow-on studies that use this dataset for research. That said, I was somewhat disappointed at the level of insight the results conveyed. It is not clear what we have learned beyond a statistical comparison of fits. The results are presented as dense tables of statistics (even the figures are graphical representations of statistical tables) where fits are classified as better or worse than others, but with little or no discussion or interpretation of the underlying biogeosciences.

*Reply 1: We thank the reviewer for these comments and by the reorganization of results and discussions, and also adding discussion, we have brought more interesting and clear message. We revised also the Introduction section to frame our work in more broad and relevant questions and dedicated space to discussion about how this work is relevant for answering ecological questions.*

Comment 2: The manuscript would clearly benefit from a more descriptive comparison of modeled vs. data. For example, I would suggest presenting Figures B1 and B2 in the main text. Perhaps see Mahecha et al. for ideas on how to gain more insight from comparisons of models and observations. Mahecha, M. D. et al. Comparing observations and process-based simulations of biosphere-atmosphere exchanges on multiple timescales. J. Geophys. Res. 115, G02003 (2010).

*Reply 2: We thank Referee #3 for the suggestion, but we think that the manuscript would lose focus when including a too detailed site-by-site analysis as was presented in Mahecha et al. However we have restructured and improved the presentation of the results and discussion sections and we have incorporated figures B1 and B2 (currently Fig. 3a and 3b) in the main text as suggested by the reviewer.*

Comment 3: One important note is that GPP and RE are modeled. From the methods it appeared that gap-filled data were also included in the fitted data. Some discussion on comparing models with modeled data is merited.

*Reply 3: It is correct that GPP and RE are not direct measurements but are derived using models where model parameters are estimated in short temporal moving windows. To acknowledge this source of uncertainty we employed GPP and RE estimates from two independent flux partitioning methods. The first extrapolates daytime ecosystem respiration using fitted relationships on the basis of nighttime data (where RE is measured due to the absence of GPP) whereas the second uses daytime NEE and an hyperbolic light response curve to derive GPP and RE. Both methods yield highly consistent results. The difference between flux partitioning methods turns out to be even smaller than the spread across ML algorithms. Regarding the fact that data are gap-filled, we filtered the data and periods with more than 20% of gap filled data with low confidence were not used in upscaling. As such the influence of gap filling was minimized. Restricting the training data set of the ML methods to periods with 100% of measured fluxes is impossible because almost no data would be available at the time resolution used. It is also important to consider that the gap filling algorithm utilizes highly localized and site specific relationships between fluxes and meteorological*

*conditions (MDS method, Reichstein et al 2005), while the ML cross-validation presented in the manuscript are based only on data from other sites. We have inserted a paragraph were the point outlined by reviewer was discussed as follows (ln 387-395):*

**"Another common issue with eddy covariance data is the gaps generated by the data exclusion rules. Data exclusion strike strongly the nighttime period (primarily for the low turbulence condition) affecting the representativeness of the diurnal cycle, hence the quality of the averaged daily/eight days eddy-covariance fluxes, in particular $CO_2$. To reduce the risk biased estimates half hourly data gaps are filled by models. In our study NEE data were gap filled using site-specific empirical relationships between meteorological data and net $CO_2$ ecosystem exchange (the MDS method, Reichstein et al., 2005) that produce small biases when short gaps were encountered (Moffat et al., 2007). This has a limited effect in this study as only a very small percentage of high quality gap filled data are used. We also minimize the bias in estimates of gross $CO_2$ fluxes (GPP and TER) by using two different partitioning methods which yield very consistent results".**

Comment 4: The authors briefly reference observational uncertainty when considering their results but it is not clear to what extent they have accounted for uncertainty. Do the models fall within the uncertainty of the observations?

*Reply 4: We did not account for the propagation of the measurement uncertainties in a formal way; however, we have added discussion on the role of uncertainties in the upscaling exercise (section 4.3 Quality of the response variable). The random uncertainty of fluxes can be thoroughly quantified but we guess it does not have a big impact because: (a) it diminishes quickly as one aggregates to daily or even eight days values, and (b) the risk of model's bias is reduced with random uncertainties. The bigger problem is related to the systematic uncertainties for which we only have some heuristic approaches to assess them (u\*, different flux partitioning methods, energy balance closure) and because, the upscaling exercise, ML parameters were across sites estimated. This issue was discussed as follows (LN 378-386):*

**"Random uncertainties of the fluxes is likely not a big issue because averaging at daily and 8 days time steps (as in this study) greatly reduces the random error (Hollinger and Richardson, 2005). Instead we hypothesize that site specific systematic uncertainties in the eddy covariance estimations (e.g. due to presence of strong advection not corrected by the standard methods) could play an important role because ML methods were trained across sites distributing uncertainties among them. Systematic uncertainties could also reduce the sensitivity of the models on the small signal explaining the comparatively poor predictive skill of ML for anomalies of eddy co-variance fluxes. We also hypothesize that the general tendency of better predictability of energy fluxes compared to carbon fluxes is at least partly related to their differences in data quality. To test these hypothesis improved ways of detecting and characterizing systematic uncertainties in eddy co-variance data are needed."**

Comment 5: The main benefit of such regression algorithms in the context of Fluxnet is scaling. It would greatly increase the impact of the paper if the authors used the trained algorithms to scale each of the fluxes to the globe. This would be relatively easy to do, and the difference between the global estimates would be much more insightful than the statistics currently presented.

*Reply 5: We agree this would add significant value in the context of scaling FLUXNET. A companion paper that uses our results as a point of departure is under preparation. It will feature global estimates as well as wall-to-wall maps.*

Comment 6: The manuscript would benefit from revisions for the correct use of English.
*The revised manuscript was proofread with the help of coauthor native*

Minor comments:

Line 31: "ML and setups"?

*Reply 6: It was correct in the revised manuscript as:*

**"Different ML and experimental setups".**

Comment 7: Line 41: Updated 2013 IPCC reference,

Line 44: "are equal"

Line 45: "accounted for"

Line 59: Perhaps cite Moffat et al. here, as it contains a good discussion of the relative benefits of both approaches. Moffat, A. M., Beckstein, C., Churkina, G., Mund, M. &Heimann, M. Characterization of ecosystem responses to climatic controls using artificial neural networks.Glob.Chang.Biol.16, 2737–2749 (2010).

Line 61: "generally come from"

Line 78: "The ML tools used span"

Line 113: "we removed 5%"

Line 294: "with respect to"

*Reply 7: Thank you, the point outlined in minor comments have been changed.*

Comment 8: Line 105: So gap-filled data of high confidence are being included? Some discussion on the dangers of fitting a model to modeled data might be warranted.

*Reply 8: We have added discussion on this issue in subsection 4.3*

Comment 9: Line 294-296: On what spatial and temporal scale? Daily NEE is typically not affected by external factors. The sentence reads as a result of the study but in reality it is a hypothesis you propose to explain the lack of model fit. You do not identify management influences or lagged effects.

*Reply 9: Yes. We have clarified it in the appropriate discussion section (4.2 Completeness of predictors) of the revised manuscript*

Comment 10: Line 298: How were the uncertainties in H, LE and NEE quantified? I do not see that presented anywhere. It is not clear where your claim that the uncertainties were larger comes from.

*Reply 10: We did not account for the measurement uncertainties in a formal way.*

Comment 11: Line 340: This sentence is not clear

*Reply 11: Sorry for the misunderstanding. The sentence was removed and the discussion reorganized.*

**Anonymous Referee #4**

*Dear Referee,*
*Thanks very much for providing detailed comments to our work. Please find enclosed the responses to all comments point-by-point.*

Comment 1: General comments:
1) Good paper, but English can significantly be improved. I added the reviewed manuscript with a lot of examples for improvement. Please take care of this action.
*Reply 1: We thank the reviewer for his/her comments. We have improved the English in the revised manuscript by the help of a native speaker and following, when possible, the suggestion by the reviewer as proposed in the supplementary material.*

Comment 2: 2) Use the present tense wherever possible.
*Reply 2: Thank you for suggestion. We have been open to this suggestion: the use the past tense was requested by the managing editor but we have changed where needed.*

Comment 3: 3) Scientifically I have no comments on this paper. Its thesis is sound and the argumentation as well.
Specific comments:
1) Page 5 line 83-84: According to me, VI's are only partially descriptive for vegetation state! Please comment and discuss on my statement.
*Reply 3: Yes, we agree with the reviewer. We have clarified this point in the revised manuscript.*

Comment 4: 2) Page 8, line 138: Give references for the QA/QC standard procedure for flux post-processing.
*Reply 4: Thanks, we have added the reference Papale et al., (2006).*

Comment 5: 3) Page 8, line 178: FPAR is an erroneous acronym for fAPAR. Please correct in the manuscript.
*Reply 5: Thank you, we have corrected it.*

Comment 6: 4) Page 9, line 189: Why was the Maximum Value Composite criterion (MVC) not used? Please explain.
*Reply 6: We have used different MODIS product, each one having specific composite method. The composite methods have been explained in the reference papers of the MODIS products (they are reported in the manuscript text) . About the point outlined by the referee, firstly we have filtered the good quality data on the basis of the MODIS quality check layer, then we have extracted the mean value of a $3 \times 3 km^2$ area centered on the tower location to reduce the effect of geolocation error and to better representing the eddy covariance footprint area, as reported in Xiao et al. (Xiao, J. ,et al: A continuous measure of gross primary production for the conterminous United States derived from MODIS and AmeriFlux data, Remote Sens Environ, 114, 576–591, doi: 10.1016/j.rse.2009.10.013, 2010). We have clarified this adding the following sentence in the revised manuscript (ln 123-124).*
**"We used MODIS cutouts of 3×3 km pixels centered on each tower to reduce the effect of geolocation error and to better representing the eddy covariance footprint area (Xiao et al., 2008; Yang et al., 2007)"**

Comment 7: 5) Page 9, line 192-193: A 16 days composite? What criterion was used for this multitemporal composite?

*Reply 7: This is the standard MODIS composition system for the product MOD13A2 as described in Huete et al (2002).*

Comment 8: 6) Page 9, line 199: VPD? Define VPD please. How was it calculated?

*Reply 8: We have define the acronym VPD (vapor pressure deficit) in the revised manuscript.*

Comment 9: 7) Page 9, line 201-202: ERA-Interim dataset? Give references for this dataset.

*Reply 9: We currently have used as reference "Dee, D. P., et al.: The ERA-Interim reanalysis: configuration and performance of the data assimilation system, Q.J.R. Meteorol Soc, 137, 553–597, doi: 10.1002/qj.828, 2011". To clarify this we have moved (Dee et al., 2011) right after 'ERA-Interim'*

Comment 10: 8) Page 17, line 387: predictive skill. This is a rather nonsensical expression, rather use predictive capacity or capability.

*Reply 10: Thank you, we have changed some expression with other among the ones proposed by the reviewer.*

Comment 11: 9) Page 18, Line 411: Individual ML methods also exhibited higher skill than... What does skill mean here? Unclear to me. Please also note the supplement to this comment:

*Reply 11: Thank you. We are referring to the predictive capacity. We have used another expression from the ones proposed by the reviewer.*

*List of the relevant changes*

*Dear Editor and Reviewers,*

*Here the list of the most relevant changes in the revised manuscript:*

a) *Objectives were rephrased and structured to be more attractive for a broader audience.*

b) *Results and discussion were splitted in two distinct sections because the same scientific topic was determinant for different facets of results (e.g the missing of import drivers affect both the ranking of bad/well predicted fluxes, the gradient of well/bad predicted plant functional type or climate zone, the difference between experimental setup).*

c) *We have moved technical details (e.g. the ones concerning the consistency among predictions by different machine learning (ML)) in appendix.*

d) *Figures in appendix were moved in the main text of Results section.*

e) *We have added relevant discussions concerning the effects of the next aspects on the ML performance: quality of the target drivers; uncertainty of the measured fluxes; gap filling of the net ecosystem exchange (NEE); the partitioning of gross $CO_2$ fluxes.*

f) *Discussions have been focused mainly on limitations of the fluxes empirical up scaling by ML. The new discussion section has been organized looking at the the following topics: a) differences between the two experimental setup (RS and RS+METEO); b) Completeness of predictors (or lack of important predictors); c) Quality of the explained variables and d) data quantity and representativeness of ecological condition and seasonal periods.*

g) *The entire manuscript was revised for the English by the help of coauthor native speaker.*

[revised manuscript text omitted]